# Revisiting In-context Learning Inference Circuit in Large Language Models

**Hakaze Cho**[1,☆]      **Mariko Kato**[1]      **Yoshihiro Sakai**[1]      **Naoya Inoue**[1,2]
[1]Japan Advanced Institute of Science and Technology     [2]RIKEN
[☆]Primary Contributor, Correspondence to: `yfzhao@jaist.ac.jp`

## ABSTRACT

In-context Learning (ICL) is an emerging few-shot learning paradigm on Language Models (LMs) with inner mechanisms un-explored. There are already existing works describing the inner processing of ICL, while they struggle to capture all the inference phenomena in large language models. Therefore, this paper proposes a comprehensive circuit to model the inference dynamics and try to explain the observed phenomena of ICL. In detail, we divide ICL inference into 3 major operations: **(1) Input Text Encode**: LMs encode every input text (in the demonstrations and queries) into linear representation in the hidden states with sufficient information to solve ICL tasks. **(2) Semantics Merge**: LMs merge the encoded representations of demonstrations with their corresponding label tokens to produce joint representations of labels and demonstrations. **(3) Feature Retrieval and Copy**: LMs search the joint representations of demonstrations similar to the query representation on a task subspace, and copy the searched representations into the query. Then, language model heads capture these copied label representations to a certain extent and decode them into predicted labels. Through careful measurements, the proposed inference circuit successfully captures and unifies many fragmented phenomena observed during the ICL process, making it a comprehensive and practical explanation of the ICL inference process. Moreover, ablation analysis by disabling the proposed steps seriously damages the ICL performance, suggesting the proposed inference circuit is a dominating mechanism. Additionally, we confirm and list some bypass mechanisms that solve ICL tasks in parallel with the proposed circuit.

## 1 INTRODUCTION

**In-Context Learning** (ICL) (Radford et al., 2019; Dong et al., 2022) is an emerging few-shot learning paradigm: given the *demonstrations* $\{(x_i, y_i)\}_{i=1}^k$ consisting of [input text]-[label token] pairs and a *query* $x_q$, **Language Models** (LMs) take the sequence $[x_1][s_1][y_1] \ldots [x_k][s_k][y_k][x_q][s_q]$[1] (Fig. 1) as input and then predicts the label for $x_q$ by causal language modeling operation. Typically, the label tokens $y_i$ are preceded by and also predicted on *forerunner tokens* $s_i$ (e.g., the colon in "Label: "). ICL has aroused widespread interest, but its underlying mechanism is still unclear.

There have been theoretical or empirical trials to characterize and explain the inference process of ICL (Xie et al., 2021; Dai et al., 2023; Wang et al., 2023; Han et al., 2023a; Jeon et al., 2024; Zheng et al., 2024). However, to capture all the operating dynamics and many fragmented observed interesting phenomena of ICL in **Large Language Models**[2] (LLMs), a more comprehensive characterization is still necessary. Therefore, this paper tries to propose a unified inference circuit and measures various properties in LLMs for a conformation to the observed ICL phenomena.

In detail, as shown in Fig. 1, we decompose ICL dynamics into 3 atomic operations on Transformer layers. **Step 1: INPUT TEXT ENCODE**: LMs encode each input text $x_i$ into linear representations in the hidden state of its corresponding forerunner token $s_i$. **Step 2: SEMANTICS MERGE**: For

---

[1]In this paper, we denote *tokenization* as [·], and *token concatenating* as [·][·].

[2]*Large* refers to scaled LMs trained by natural language data, such as Llama 3 (AI@Meta, 2024), contrast to simplified work that uses simple models trained and test on well-embedded input in toy models.

demonstrations, LMs merge the encoded representations of $s_i$ with the hidden state of the corresponding label tokens $y_i$. **Step 3: FEATURE RETRIEVAL AND COPY**: LMs retrieve the similar semantics-merged label representations $y_{1:k}$ (from Step 2) to the query representation $s_q$ in a task-relevant subspace, then copy them into the query's forerunner token representation. Finally, LM heads predict the label for $x_q$ using the label-attached query representation $s_q$. Steps 2 and 3 form a typical induction circuit, which is a key mechanism of ICL but only examined in synthetic scenarios (Elhage et al., 2021; Singh et al., 2024b; Reddy, 2024).

We empirically find evidence for the existence of each proposed step in LLMs, and conduct more fine-grained measurements to gain insights into some phenomena observed in ICL scenarios, such as (1) *positional bias*: the prediction is more influenced by the latter demonstration (Zhao et al., 2021), (2) *noise robustness*: the prediction is not easy to be affected by demonstrations with wrong (*noisy*) labels (Min et al., 2022), while larger models are less robust to label noise (Wei et al., 2023), and (3) *demonstration saturation*: the accuracy improvements plateau when sufficient demonstrations are given (Agarwal et al., 2024; Bertsch et al., 2024), etc. (discussed in §5.3). Moreover, we find multiple bypass mechanisms for ICL conducted by residual connections, while the 3-phase dynamics remain dominant.

Figure 1: The 3-phase inference diagram of ICL. **Step 1**: LMs encode every input text into representations, **Step 2**: LMs merge the encoded text representations of demonstrations with their corresponding label semantics, **Step 3**: LMs retrieve merged label-text representations similar to the encoded query, and copy the retrieved representations into the query representation.

**Our contributions can be summarized as:** (1) We propose a comprehensive 3-step inference circuit to characterize the inference process of ICL, and find empirical evidence of their existence in LLMs. (2) We conduct careful measurements for each inference step and successfully capture a large number of interesting phenomena observed in ICL, which enhances the practicality of the proposed circuit. (3) Our ablation analysis suggests that the proposed circuit dominates, but some bypass mechanisms exist in parallel to perform ICL, and we introduce some of these bypasses along with their empirical existence evidence.

## 2 PREPARATION

### 2.1 BACKGROUND & RELATED WORKS

**In-context Learning.** Discovered by Radford et al. (2019), ICL is an emerging few-shot learning paradigm with only feed-forward calculation in LMs. Given demonstrations $\{(x_i, y_i)\}_{i=1}^k$ composed of input-label pairs and a query $x_q$, typical ICL creates an input $[x_1][s_1][y_1] \ldots [x_k][s_k][y_k][x_q][s_q]$, with some structural connectors (e.g. "Label: ") including forerunner token $s_i$ (e.g. ": "), as shown in Fig. 1. LMs receive such inputs and return the next token distribution, where the label token with the highest likelihood is chosen as the prediction. Explaining the principle of ICL is an open question, although there have been some efforts on **the relationship between ICL capacity and pre-training data** (Li & Qiu, 2023; Singh et al., 2024b;a; Gu et al., 2023; Han et al., 2023b; Chan et al., 2022), the **feature attribution of inputs** (Min et al., 2022; Yoo et al., 2022; Pan, 2023; Kossen et al., 2024), and **reduction to simpler processes** (Zhang et al., 2023; Dai et al., 2023; Xie et al., 2021; Han et al., 2023a). However, a comprehensive and unified explanation on real-world LMs is still needed to capture the operating dynamics of ICL.

**Induction Circuit.** Introduced by Elhage et al. (2021), an induction circuit is a pair of two cooperating attention heads from two transformer layers, where the "previous token head" writes information about the previous token to each token, and the "induction head" uses the wroten information to identify a token that should follow each token. Concretely, such a function is implemented by two atomic operations in attention calculations: (1) copy the representation of the previous token [A] to the next token [B], and (2) retrieve and copy similar representations on [B] (copied from [A]) to the current token [A']. Concisely, it performs inference in the form of $[A][B] \ldots [A'] \Rightarrow [B]$, which is similar to the ICL diagram. Therefore, the induction circuit has been widely used to explain the in-

ference dynamics of ICL (Wang et al., 2023) and the emergence of ICL during pre-training (Olsson et al., 2022; Reddy, 2024; Singh et al., 2024b). Despite their valuable insights, these experiments rely on a synthetic setting: using simplified models and well-embedded (linearly separable) inputs, which differs from the practical ICL scenario using real-world LMs with many layers and complicated inputs: we believe these synthetic works indicate the *potential* of Transformers or a locally optimal behavior, and can not demonstrate that their observation exists in scaled LLMs trained on wild data. So, in this paper, we try to bridge the gap between the synthetic and real-world settings with more detailed observations for the inference of ICL on real-world LLMs.

## 2.2 EXPERIMENT SETTINGS

**Models.** We mainly conduct experiments on 4 modern LLMs: Llama 3 (8B, 70B) (AI@Meta, 2024), and Falcon (7B, 40B) (Almazrouei et al., 2023). Unless specified, we report the results on Llama 3 70B, since its deep and narrow structure (80 layers, 64 heads) makes it easier to show hierarchical inference dynamics (discussed in §5.2). The results of other models can be found in Appendix H.2.

**Datasets.** We build ICL-formed test inputs from 6 real-world sentence classification datasets, and unless specified, we report the average results on them: SST-2 (Socher et al., 2013), MR (Pang & Lee, 2005), Financial Phrasebank (Malo et al., 2014), SST-5 (Socher et al., 2013), TREC (Li & Roth, 2002; Hovy et al., 2001), and AGNews (Zhang et al., 2015).

**Others.** Unless specified, we use $k = 4$ demonstrations in ICL inputs. For each dataset, we randomly sample $512$ test data points and assign one fixed demonstration sequence for each test sample to form a test input. About the prompt templates, etc., please refer to Appendix A.1.

## 3 STEP 1, INPUT TEXT ENCODE: SEMANTICS ENCODING AS LINEAR REPRESENTATIONS IN HIDDEN STATES

This section mainly confirms that LMs construct task-relevant and linearly separable semantic representations for every input text (demonstrations and queries) in the hidden states. Such linear representations are an important foundation for explaining the dynamics of ICL based on induction heads, since attention-based feature retrieval, a key mechanism of induction heads, can be easily done on linear representations. Current successful studies on simplified models and inputs (Chan et al., 2022; Reddy, 2024; Singh et al., 2024b) also assume the existence of such linear representations. Moreover, we confirm some interesting properties of the encoded input text representations: (1) It is based on the capacity in the model weights and can be enhanced by demonstrations in context. (2) The similarity of representations is biased towards the encoding target's position.

## 3.1 LLMS ENCODE INPUT TEXT ON FORERUNNER TOKENS IN HIDDEN STATES

We first study the existence of input text encoding in hidden states and then explain their linear separability and task relevance in §3.2. For each text-label pair $(x_t, y_t)$ (*encoding target*) sampled from the datasets, we prepend them with $k$ demonstrations, resulting in augmentated (by label $[y_t]$) ICL-styled inputs $[x_1][s_1][y_1]\ldots[x_k][s_k][y_k][x_t][s_t][y_t]$. These inputs are then fed into an LM to extract the hidden states of a specific token in $[x_t][s_t][y_t]$ from each layer, serving as the ICL inner representations. To assess the quality of these representations as sentence representations, we use the sentence embedding of $x_t$ encoded by BGE M3 (Chen et al., 2024), a SotA encoder-only Transformer, as a *reference representation* and then calculate the *mutual nearest-neighbor kernel alignment*[3] (Huh et al., 2024) between these representations. See Appendix A.1 and A.2 for details.

**Forerunner Tokens Encode Input Text Representations.** We plot the kernel alignment on the 3 types of tokens in Fig. 2 (Left), where the forerunner token, while often overlooked in previous work, produces the best input text encoding, emerging in the early phase (layers 0-28) of the inference process, and keeping a high level to the end of inference. Interestingly, hidden states of label words are not satisfactory input text representations even with a high background value (the result at layer

---

[3]Intuitively, kernel alignment measures similarity between two representations toward the same datasets, and according to Huh et al. (2024), a higher cross-model kernel alignment usually means a better representation.

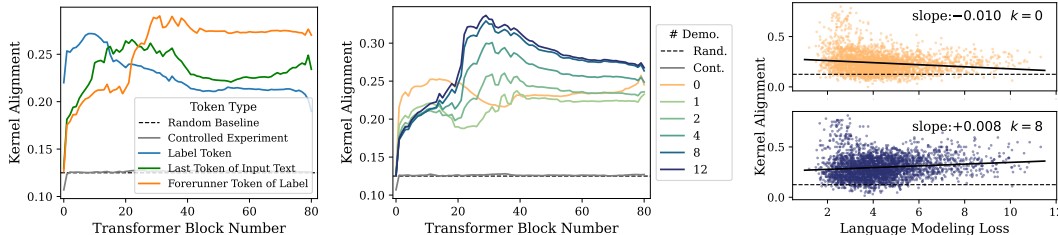

Figure 2: Input text encoding magnitudes (metricized by kernel alignment with feature encoded by an encoder-structured model) of hidden states in various layers in ICL scenario (The controlled experiments are results between current 6 datasets and TEE (Mohammad et al., 2018)). **Left**: Encoding magnitudes on hidden states from various types of token. **Middle**[4]: Encoding magnitudes with different $k$ on the forerunner tokens. **Right**: Encoding magnitudes in layer 24 of Llama 3 70B against the causal language modeling loss of the input text with (upper) $k = 0$ and (lower) $k = 8$.

0, refer to Appendix A.2.1), which is a critical supplement to previous work which suggests the label tokens are pivots for collecting the information of demontrations (Wang et al., 2023).

**Input Text Encoding is Enhanced by Demonstrations.** We investigate the influence of contextual information on input text encoding by repeating the experiments with different $k$. As shown in Fig. 2 (Middle), when the demonstrations increase, kernel alignment is enhanced, which is counterintuitive since longer preceding texts are more likely to confuse encoding targets. Such findings indicate that LMs (1) utilize contextual information to enhance the input text encoding and (2) correctly segment different demonstrations (detailed operation discussed in Appendix C).

**Perplexed Texts are Encoded Worse.** We investigate the correlation between kernel alignment and the perplexity of encoding targets with various $k$. Fig. 2 (Right, upper) shows a negative correlation for $k = 0$, that is, LMs generate poorer encodings for more perplexed input text when no demonstrations are given, which can be identified as an **I**n-**w**eight **L**earning (IWL) property of the inner text encoding. While, when demonstrations are given in context (Fig. 2 (Right, lower)), the negative correlation disappears, which suggests that LMs effectively encode more complex samples with the help of demonstrations in context. More discussion about the correlation between classification performance and perplexity is in Appendix F.

The above findings suggest that the inner text encoding is a hybrid process of ICL and IWL: Basic encoding capability presents from LMs weights, and is enhanced by demonstrations in context, which can be a clue to how demonstrations help ICL. Moreover, we are about to illustrate that these encodings are sufficiently informative for ICL tasks and linearly separable, which meets the presumption of simplified models on the linear and well-embedded input features.

### 3.2 INPUT TEXT ENCODING IS LINEAR AND TASK-RELEVANT BUT POSITION-BIASED

**Input Text Encoding is Linear Separable and Task-relevant.** We train a centroid classifier on hold-out 256 input samples (Cho et al., 2024), using the hidden states of a specific token in $[x_t][s_t]$[5] from each layer and then predict the label $y_t$ (see Appendix A.3 for details). The results are shown in Fig. 3, where considerably high classification accuracy of the forerunner token suggests the high linear separabilities of the hidden states in the task-semantic-relevant subspaces since the centroid classifier is linear. In addition, a similar emerging trend in accuracy and kernel alignment confirms the reliability of the kernel alignment measurement.

**Input Text Encoding is Biased towards Position.** Ideally, the inner representations of similar queries should be highly similar regardless of their position in ICL inputs to support attention-

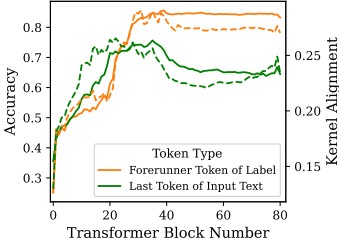

Figure 3: Test results of centroid classifier trained on ICL hidden states. **Solid**: Centroid classification accuracy, **Dotted**: Kernel alignment.

---

[4]Experiments of Fig. 2 (Middle) on Llama 3 70B do not involve results on AGNews.
[5]We skip the experiments on label tokens because of the leakage of ground-truth label information.

based operations for classification. To verify this, for each encoding target, we extract the hidden states of forerunner tokens with various numbers of preceding demonstrations, and then calculate the cosine similarity between all possible pairs of the hidden states for the same target or different targets. As shown in Fig. 4, although the overall similarities on the same target are higher than on the different targets, they are both especially higher when their positions are close to each other. As to be discussed in §5.3, such positional similarity bias may lead to one flaw: Demonstrations closer to the query have stronger impacts on ICL (Zhao et al., 2021; Lu et al., 2022; Chang & Jia, 2023; Guo et al., 2024). The principle of such bias is discussed in Appendix C.

## 4 INDUCTION CIRCUITS IN LARGE LANGUAGE MODELS

This section mainly shows how LMs utilize the encoded linear text representations in induction circuits with a typical 2-step form (Singh et al., 2024b): **Forerunner Token Heads** merge the demonstration text representations in the forerunner tokens into their corresponding label tokens with a selectivity regarding the compatibility of demonstrations and label semantics. **Induction Heads** copy the information in the label representations similar to the query representation back to the query, which is conducted on task-specific subspaces, enabling LMs to solve multiple tasks by multiplexing hidden spaces.

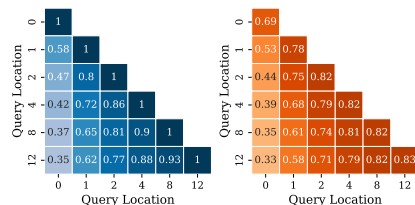

Figure 4: The similarities of ICL hidden states in different positions on layer 24 between **Left**: the same queries, **Right**: two different queries (on SST-2).

### 4.1 STEP 2, FORERUNNER TOKEN HEAD: COPY FROM TEXT FEATURE TO LABEL TOKEN

This subsection mainly examines and measures the forerunner token heads, which copy the information in the forerunner token into label tokens. We focus on how the representations in these tokens are merged, especially when the semantics of demonstrations and corresponding labels are disjoint, to explain the robustness of ICL against wrong labels present in the demonstrations.

**Text Representations are Copied to Label Tokens.** To confirm the existence of the representation copy process, we start by calculating the kernel alignment between the hidden state of forerunner token $[s_t]$ at layer $l$ (the copy source) and that of label token $[y_t]$ at layer $(l+1)$ (the copy target). To suppress the high background values caused by the semantics of labels (refer to Appendix A.2.1), we use abstract label tokens $\{$ "A", "B", "C", ... $\}$ instead of the original label tokens. The results are shown in Fig. 5 (Left), where the kernel alignment between the hidden states of the label token and the forerunner token gradually increases and then bumps up after the encoding magnitude (described in §3) in the forerunner token finished ascending. Such a phenomenon indicates that the hidden states of the input text representation encoded in the forerunner tokens are merged into their label tokens, suggesting the existence of copy processing from the forerunner token to the label token.

**Text Representations are Copied without Selectivity.** For each attention head, we extract the attention score[6] $\alpha_{s_t \rightarrow y_t}$ from the forerunner token $[s_t]$ (as attention key) to the corresponding label token $[y_t]$ (as attention query). We then mark the head with $\alpha_{s_t \rightarrow y_t} \geqslant 5/n_t$ ($n_t$: the length of tokens before $[y_t]$) as a Forerunner Token Head and count them in each layer. The results are shown in Fig. 5 (Middle, "Correct Label"), where the peak matches the copy period in Fig. 5 (Left). Moreover, to investigate the influence of the correctness of label tokens, we replace $[y_t]$ with a wrong label token[7], where the results in Fig. 5 (Middle, "Wrong Label") are almost identical to the correct-label setting, suggesting that the forerunner token heads don't show selectivity toward the semantic consistency between input text and labels, and simply merge the input text representations into the label tokens. Furthermore, we find (Appendix D) that the copy processing is *inherent*: During the copy processing, the preceding tokens are copied to the subsequent tokens, regardless of whether these copied tokens are forerunner tokens, which indicates that such copy processing is a universal inference behavior established during pre-training, rather than being evoked by the special tokens in the in-context learning (ICL) input, while still aiding ICL process (discussed in §5.1).

---

[6]This paper use notation $\alpha_{K \rightarrow Q}$ to denote attention score with attention query $Q$ and attention key $K$, which is same with the flow of information.

[7]For example, for a label space of "positive" and "negative", if $[y_t]$ is "positive", we replace it to "negative".

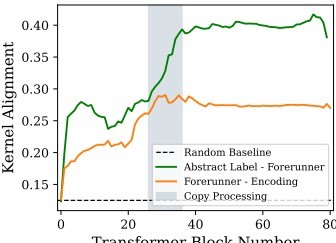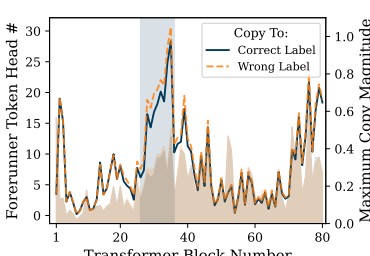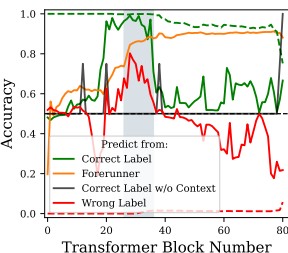

Figure 5: Hidden states copy magnitude from forerunner tokens to label tokens against layers. **Left**: Kernel alignment between the forerunner token (the copy source) and the abstract label token of the next layer (the copy target). **Middle**: **Curves**: The count of marked forerunner token heads with correct and wrong labels; **Colored Areas**: The maximum attention scores from forerunner token to query (*copy magnitude*) with correct and wrong labels (detailed attention head statistical data is in Appendix H.1). **Right**: Centroid classifier results predicted on the hidden states of correct and wrong *label tokens*, on SST-2 and MR. **Solid**: Predicted by classifiers $\mathcal{C}_s$ trained on hidden states of *forerunner tokens*. **Dotted**: Predicted by classifiers $\mathcal{C}_y$ trained on hidden states of *label tokens*.

**Hidden States of Label Tokens are Joint Representations of Text Encodings and Label Semantics.** Given the findings above, we probe the content of hidden states of label tokens $[y_t]$, i.e., how the copied text representation interacts with the original label semantics. We first train two centroid classifiers to predict the corresponding label $y_t$: (1) $\mathcal{C}_s$ trained on the hidden states of *forerunner tokens* $[s_t]$ and (2) $\mathcal{C}_y$ trained on the hidden states of *label tokens* $[y_t]$. To check whether the label tokens include the information of forerunner tokens, we use $\mathcal{C}_s$ to predict the label on the hidden state of label token $[y_t]$ in Fig. 5 (Right, solid). It shows that, during the copy processing, high classification accuracies can be achieved both on the correct label tokens and wrong label tokens, suggesting that the text features in the forerunner tokens can be partly and linearly detected in the label tokens. Moreover, results using $\mathcal{C}_y$ (dotted line) show extreme results, suggesting the label information remains in the label token. So, we can conclude that: Hidden states of label tokens are joint representations of label semantics and text representations. Moreover, interestingly, the accuracies from $\mathcal{C}_s$ shown in Fig. 5 (Right) decline after layer 35, suggesting that the information sharing between the forerunner tokens and the label tokens ends in later layers, which aligns with the results in Fig. 5 (Middle).

**Label Denoising is Conducted on the Overlap of Label Semantics and Text Representations.** Notice that in Fig. 5 (Right, solid), compared to the typical results predicted on the forerunner tokens, accuracies are improved on the correct label and suppressed on the wrong label, which suggests that information consistent with the label semantics is easier to be enhanced by the label tokens and vice versa, showing a feature selectivity on the consistency between the text representations and the label semantics. Given the observation that the information on label semantics and text features can be extracted separately and linearly, we can confirm that these two kinds of information are located in different sub-spaces of the hidden states, and *linearly* merged by the attention operation of forerunner token heads. Moreover, given the fact that there is no selectivity is observed in the copy behavior of the forerunner token heads (Fig. 5 (Middle)), it is intuitive that the feature selectivity shown in Fig. 5 (Right) comes from the arithmetical interaction of feature vectors on the *overlap* of sub-spaces between the label semantics and text features, making ICL stable against label noise (Min et al., 2022). Moreover, as mentioned by Wei et al. (2023), large models show poorer stability against label noise, and we infer that the larger hidden dimensions in larger models lower the overlap between the sub-spaces of label semantics and text representations to reduce the interaction.

## 4.2 STEP 3, INDUCTION HEAD: FEATURE RETRIEVAL ON TASK SUBSPACE

This subsection examines the existence of the induction heads, which retrieve label token features similar to the queries' forerunner token feature, and copy the retrieved features back to the query. We claim the necessity of multi-head attention in this process: Correct feature retrieval can only be conducted on the subspace of the hidden space, which is captured by some attention heads.

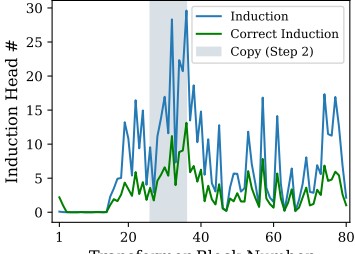 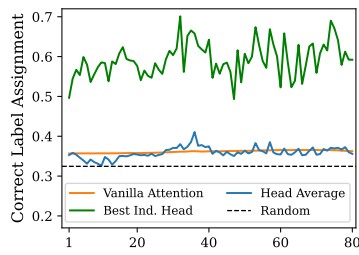 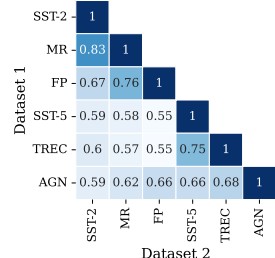

Figure 6: Measurements for induction heads. **Left**: The count of marked induction heads and correct induction heads against layers. **Middle**: The correctness of attention assignment (the sum of attention scores from query's forerunner token towards correct label tokens normalized by attention scores towards all the label tokens) extracted from the following attention calculations: **Vanilla Attention:** attention scores directly calculated on full dimensionality. **Head Average:** the averaged attention scores among all the heads. **Best Ind. Head:** the scores of attention head with the most correctness. **Right**: The correct induction heads overlap of all dataset pairs.

**Induction is Correct in Minority Subspaces.** Similar to Fig. 5 (Middle), we mark (1) the attention heads with the sum of attention scores from all the label tokens in the demonstrations $[y_1], \ldots, [y_k]$ (as attention keys) to the query's forerunner token $[s_q]$ (as attention query) of more than $5k/n_t$ as *Induction Heads*, and (2) attention heads with the sum of scores from all the correct label tokens more than $5k/|\mathbb{Y}|n_t$ as *Correct Induction Heads* ($\mathbb{Y}$ is the label space). We show the number of both kinds of induction heads in Fig. 6 (Left, detailed head statistics in Appendix H.1), where an unimodal pattern is observed later than the copy processing of Step 2. Moreover, more than half of the induction heads are not correct ones, suggesting that task-specific feature similarity can only be caught on some *induction subspaces* (defined by low-rank transition matrix $W_Q^{h\top} W_K^h$ of correct induction head $h$). We enhance this claim in Fig. 6 (Middle) (details in Appendix A.4), where both vanilla attention (without transformation and head split) and attention scores averaged among all heads show low assignment on correct label tokens, while some heads show considerable correctness. Considering the average value, the majority of attention heads almost randomly copy label token information to the query, causing the final prediction biased to the frequency of labels present in the demonstrations (Zhao et al., 2021). As the reason, we infer that the hidden states are sufficient (Fig. 3) but not minimum for ICL, where redundant information interferes with the operation of induction heads.

**Some Induction Subspaces are Task-specific.** We check if different tasks share the same induction subspaces based on the overlap of the correct induction heads across different datasets. Given $n_\mathcal{D}(h)$ as the number of times $h$ is marked as correct induction head on dataset $\mathcal{D}$, the overlap rate $S$ is defined as:

$$S(\mathcal{D}_1, \mathcal{D}_2) = \frac{2 \sum_{\forall h} \min[n_{\mathcal{D}_1}(h), n_{\mathcal{D}_2}(h)]}{\sum_{\forall h} n_{\mathcal{D}_1}(h) + n_{\mathcal{D}_2}(h)}.$$

The results are shown in Fig. 6 (Right), where: (1) A significant overlap of induction heads indicates that a part of correct induction heads is *inherent* in the model, built by the pre-training process (Reddy, 2024; Singh et al., 2024b). (2) Such overlap is not fully observed, suggesting that some induction subspaces are *task-specific*:

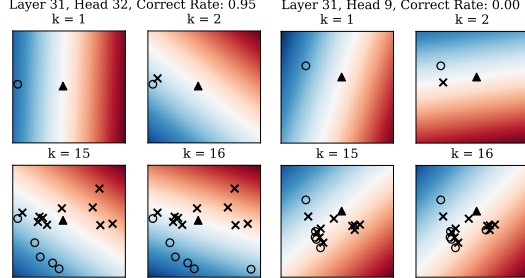

Figure 7: Label representations of $k$ demonstrations visualized on **Left 4**: correct and **Right 4**: wrong induction head, on one sample of SST-2 (see Appendix H.3). ○: "positive" label, ×: "negative" label, ▲: zero vector. Color: Radial density of each categories, to estimate the attention assigned to query, negative to positive (cartography: Appendix A.5).

Input texts evoke task-specific induction heads, enabling the anisotropy multiplex of different subspaces in the hidden spaces to transmit relevant information for various tasks. Therefore, we can analogize ICL as an implicit end-to-end multi-task learning with hidden state multiplexing since multi-task learning also utilizes various task heads on common bottom network layers and informative hidden states (Zhang & Yang, 2021).

**Demonstrations Saturate on Induction Subspace.** We visualize the demonstrations' label token representations mapped on the induction subspaces by the transition matrix $W_Q^{h\top} W_K^h$ and principal component analysis in Fig. 7, indicating: (1) Compared to correct induction heads (Left 4), wrong induction heads (Right 4) are easier to map label representations linear-inseparably. (2) In the early stage of demonstration feeding ($k = 1 \to 2$), when a new demonstration is given, the morphology of attention assignment towards query updates significantly (shown as background color in Fig. 7, estimated by the radial density of various labels; see §A.5 for details), while in the late stage ($k = 15 \to 16$), attention assignment morphology is stable. This can explain the demonstration saturation (Agarwal et al., 2024; Bertsch et al., 2024): ICL performance is submodular against the number of demonstrations. Intuitively, since demonstrations follow a prior distribution, representation of a new demonstration is likely to be located within the closure of existing demonstrations, making it less diverse to contribute to the attention assignment in the induction subspace.

## 5   PUTTING THINGS TOGETHER

So far, we have revealed the existence of the circuit with 3 steps, organized by the sequential inference process among Transformer layers. In this section, we find that the circuit is dominant in the ICL inference, while some bypass mechanisms activated by residual connection assist ICL inference. Moreover, a series of phenomena observed in ICL is successfully explained by the circuit.

### 5.1   ABLATION ANALYSIS

To (1) examine the causality between the attention connection specified by the 3 steps and ICL inference, and (2) demonstrate that our 3-phase circuit dominates or at least participates in ICL process, we disconnect the related attention connection of each step in the proposed circuit (see Appendix A.7 for details), and test the accuracies without such connections, as shown in Table 1. The results indicate that, compared to the **controlled results**, where trivial connections are removed, the accuracy of ICL significantly decreases

Table 1: Accuracy variation (%) with each inference step ablated on Llama 3 8B. Small numbers are **controlled results** (mean ± std) of randomly ablating equivalent amounts of connections. Ablations are applied from the bottom to the top layers, and results with various ablated layers are reported (detailed settings: Appendix A.7).

| # | **Attention Disconnected** Key → Query | **Affected Layers Ratio** (from layer 1) | | | |
|---|---|---|---|---|---|
| | | 25% | 50% | 75% | 100% |
| 1 | **None** (4-shot baseline) | ±0 (Acc. 68.55) | | | |
| | *– Step1: Input Text Encode –* | | | | |
| 2 | **Demo. Texts** $x_i$ → **Forerunner** $s_i$ | −4.98 
 −0.89 ± 0.00 | −15.82 
 −1.19 ± 0.02 | −23.43 
 −3.29 ± 1.87 | −30.60 
 −1.61 ± 0.01 |
| 3 | **Query Texts** $x_q$ → **Forerunner** $s_q$ | −13.87 
 −0.16 ± 0.00 | −21.10 
 −0.08 ± 0.00 | −24.74 
 −0.47 ± 0.04 | −28.38 
 −0.55 ± 0.00 |
| | *– Step2: Semantics Merge –* | | | | |
| 4 | **Demo. Forerunner** $s_i$ → **Label** $y_i$ | −2.24 
 −0.00 ± 0.00 | −3.45 
 −0.18 ± 0.00 | −3.39 
 −0.10 ± 0.04 | −3.42 
 −0.18 ± 0.01 |
| | *– Step3: Feature Retrieval & Copy –* | | | | |
| 5 | **Label** $y_i$ → **Query Forerunner** $s_q$ | −5.14 
 +0.03 ± 0.00 | −10.03 
 −0.08 ± 0.00 | −11.36 
 +0.00 ± 0.00 | −10.22 
 −0.06 ± 0.00 |
| | *Reference Value* | | | | |
| 6 | **Zero-shot** | −17.90 (Acc. 50.65) | | | |
| 7 | **Random Prediction** | −36.05 (Acc. 32.50) | | | |

when the non-trivial connections designated by the proposed circuit are ablated. This supports the existence of our circuit. However, the result doesn't fully match expectations, for example, the result without induction (line 5) should be consistent with the zero-shot inference result (line 6), since all the expected communication from demonstration to query is intercepted, but the real result is better; and the contribution of Step 1 **in later layers** are unexpectedly high, indicating the existence of some bypass mechanisms parallelly contributing to ICL accuracies.

### 5.2   BYPASS MECHANISM

Motivated by the ablation results, we believe that several mechanisms including our circuit run parallelly for ICL, since the residual connection supports complex paths among layers and attention heads. We list some possible bypasses and plan a complete enumeration as future work.

**Parallel Circuits.** Multiple 3-step circuits can execute in parallel, and one layer can assign

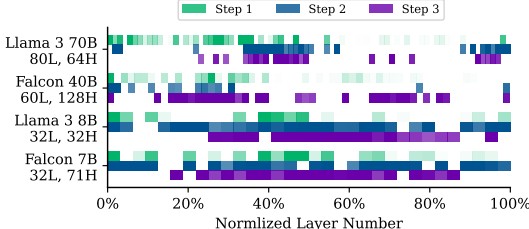

Figure 8: Dynamics and deserialization of magnitudes of proposed 3 inference steps (cartography details: Appendix A.6).

multiple inference functions to different heads, causing dispersion and deserialization as shown in Fig. 8, where a narrow (fewer heads per layer) and deep (more layers) model is more likely to generate localized inference, and vice versa.

**Direct Decoding.** Residual connection to the LM heads allows intermediate hidden states to be decoded directly. Intuitively, a shortcut from Step 1 encodings to the LM head enables ICL with zero-shot capacities, since we have confirmed that encoded representations are informative for ICL tasks (§3.2), while the decoding methods should be selected carefully (Cho et al., 2024) (e.g. with essential calibration). On the other hand, shortcuts from insufficiently encoded features may lead to meaningless information decoded by language model heads, causing prediction bias, i.e., even if no query is given, ICL still returns unbalanced results (Zhao et al., 2021) decoded from tokens of prompt template (see Appendix E for details).

Table 2: Accuracy drop with shortcut ablated.

| **Attention Disconnected** ($\forall i$) | 25% | 50% | 75% | 100% |
|---|---|---|---|---|
| **Forerunner** $s_{1:i} \rightarrow$ **Forerunner** $s_{i:q}$ | $-1.30$ $-0.00$ | $-0.75$ $-0.16$ | $-0.78$ $-0.16$ | $-1.56$ $-0.71$ |

**Shortcut Induction.** Note that a $k$-shot ICL input sequence always contains a $(k - 1)$-shot sequence, where the $k$-th forerunner token (served as the query of $(k-1)$-shot sequence) is previously processed by the $(k-1)$-shot inference. So every forerunner can directly retrieve previously processed forerunner tokens in the demonstrations to copy their induction results directly. The non-trivial result (Table 2) with all forerunner tokens disconnected from each other confirms such infer.

## 5.3 EXPLANATION TOWARDS OBSERVED ICL PHENOMENA

**Difficulty-based Demonstration Selection.** In §3.1, we find that in the zero-shot scenario, perplexed texts are harder to encode, explaining the observation of PPL-ICL (Gonen et al., 2023): Selecting demonstrations with lower perplexity can improve ICL performance. Moreover, while the demonstrations increase, LMs can encode more complex inputs with diverse information to update the attention assignment shown in Fig. 7, making it beneficial to input harder demonstrations later, which explains the ICCL (Liu et al., 2024), which build demonstrations sequence from easy to hard.

**Prediction Bias.** (1) **Contextual Bias**: As mentioned in §5.2 and shown in Appendix E, direct decoding insufficiently encoded information adds meanless logits into LM's output, causing a background prediction value even if no queries are given (named *bias*). (2) **Positional Bias**: As shown in §3.2, closer demonstrations are encoded more similarly, so label tokens near the query have more similar information to the query, causing more attention assignment in the induction processing, so that more influences on the prediction. (3) **Frequency Bias**: As shown in §4.2, in the induction, some attention heads are without correct selectivity towards labels, causing an averaged copy processing from label tokens to the query, triggering a prediction bias towards the label frequency in the demonstration, even if their contribution (absolute value of attention score on label tokens) is small. All three biases are observed by Zhao et al. (2021), and can be removed by ICL calibration methods.

**The Roles and Saturates of Demonstrations.** It is well known that demonstrations improve the performance of ICL. We decompose such performance improvement into 2 parts: (1) demonstrations help early layers encode better (§3.1), and (2) more demonstrations provide larger label token closure, enabling more accurate attention assignment (§4.2), while the volume of such closure is submodular to demonstrations, causing the saturates of ICL performance towards demonstrations.

**The Effect of Wrong Label.** It is well-known that the label noise is less harmful in ICL (Min et al., 2022) than in gradient-based learning (Zhang et al., 2021). We have explained in §4.1 that ICL implies labels denoise to stabilize ICL against label noise, while weakened by dimensionality.

## 6 CONCLUSION AND DISCUSSION

**Conclusion.** In summary, this paper restores ICL inference into 3 basic operations and confirms their existence. Fine-grained measurements are conducted to capture and explain various phenomena successfully. Moreover, ablation studies show the proposed inference circuit dominates and reveals the existence of bypass mechanisms. We hope this paper can bring new insight into ICL practice.

**The Role of Early and Later Layers.** Our framework shows: The encoding result of Step 1 can be directly used for classification with reliable decoding, and later transformer layers are not contributing to centroid classification accuracies (Fig. 3), leading to a taxonomy of *Input Encoding* for

Step 1 and *Output Preparation* for Step 2 and 3: LMs complete multi-task classification implicitly in early layers, and verbalize it by merging task-specific label semantics in later layers. Therefore, we suggest an early-exiting inference: removing some top layers and using a centroid classifier (Cho et al., 2024) to accelerate ICL as shown in Table 3 and Appendix E.

**Pre-training Possibility from Natural Language Data.** A large gap can be considered between such a delicate circuit and gradient descent pre-training on the wild data. However, we believe the wild training target contains the ICL circuit *functionally*: Based on the previous works finding trainability of ICL on simplified linear representation-label pairs (Chan et al., 2022; Reddy,

Table 3: Performance of full and layer-pruned ICL inference.

| Inference | Acc. | # Param. | Speed |
|---|---|---|---|
| Full + LM Head | 66.19% | 70.6B | 1× |
| Full + Centroid | 83.24% | 69.5B | 1.00× |
| Layer34 + LM Head | 49.29% | 32.7B | 2.16× |
| Layer34 + Centroid | **84.27%** | **31.2B** | **2.38×** |

2024; Singh et al., 2024b), we speculate that in early training step, Transformers learn to extract linear representations shown in §3 from wild data (detailed in Appendix B), serving as the training input of later layers to evoke the emergence of induction heads with the same mechanism shown in aforementioned previous works. Moreover, our conclusion of Step 3 highlights the input data requirements for the later layers: These data should activate the multiplex of hidden space, i.e., it should implicate multi-task classification with a wide distribution, which is consistent with the aforementioned previous works.

**Comparison with Previous Works.** Several prior studies have sought to interpret ICL as known processes, including implicit gradient descent (Dai et al., 2023), kernel regression (Han et al., 2023a), and implicit Bayesian inference (Xie et al., 2022), etc. However, as stated in §2.1, these approaches fall short of fully explaining the phenomena observed during the ICL inference process. For instance, gradient descent is known to be fragile against label noise (Zhang et al., 2021), leading to a misalignment when analogies are drawn to ICL, which is robust against label noise (Min et al., 2022; Liu et al., 2020). Similarly, attempts to explain ICL as kernel regression fail to account for positional bias, making them disconnected from empirical studies that have observed various inference phenomena in real-world LMs. Also, other works have employed induction circuits to explain ICL dynamics (Wang et al., 2023; Elhage et al., 2021; Olsson et al., 2022), yet significant gaps remain in aligning these explanations with empirical observations. Our work is the first to unify the fragmented conclusions from prior empirical studies (discussed in §5.3) through detailed experimental measurements. By demonstrating the alignment of these observations, we emphasize the novelty and primary contribution of our paper.

**Limitations & Open-questions.** (1) These 3 basic operations are not functionally indivisible. Ideally, mechanistic interpretability aims to reduce every operation in ICL inference to the interconnection of special attention heads to ulteriorly examine how the operating subspaces interact between steps, and reconstruct ICL behavior from a minimal set of attention heads. Also, although we show the significance of the inference circuit in the ablation analysis (§5.1), measuring the connectivity of these attention heads can also be beneficial to get more insights into the circuit. (2) The conclusions may not align with scenarios where ground-truth labels are not provided in the context in some cases, which are often referred to as in-weight learning, where significant differences or even antagonism with standard ICL have been highlighted by previous works[8] (Chan et al., 2022; Reddy, 2024), reasonably and necessarily warranting separate discussion (discussed in Appendix G). This paper explains the inference behavior of the model under ICL conditions, leaving the in-weight learning scenario for future works. (3) We only focus on classification tasks, while we believe that our findings can be applied to non-classification tasks, efforts are still needed to fill the gap.

## AUTHOR CONTRIBUTIONS

Hakaze Cho, also known as Yufeng Zhao, handled the entire workload in this paper. He provided ideas, designed / conducted experiments, collected / described data, and wrote / revised the paper.

Naoya Inoue, as the supervisor, provided valuable feedback, revisions, and financial support for this paper. M.K. and Y.S. made minor contributions that placed them on the borderline of authorship criteria: they participated in discussions and offered comments, some of which were incorporated.

---

[8]In these works, toy models even struggle to work on both inputs simultaneously, so that it is natural to consider different inference mechanism for the so-called in-weight learning.

REPRODUCIBILITY STATEMENT

The official code implementation of this paper by the author can be found at `https://github.com/hc495/ICL_Circuit`. Please follow the instructions in this GitHub repository to reproduce the experiments. If any final data or intermediate results are required, please get in touch with the first author by contact information shown in ⓘ.

ACKNOWLEDGMENTS

This work is supported by The Nakajima Foundation Start-Up Support Grant.

The authors would like to thank Mr. Lihao Liu at the Beijing Institute of Technology, and Ms. Yunpin Li at the Capital Normal University for proofreading.

The authors would like to sincerely thank the chairs and reviewers of ICLR 2025 for their thoughtful and insightful feedback on this paper, their perceptive reviews and discussions have greatly improved this work.

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

# Revisiting In-context Learning Inference Circuit in Large Language Models
# Appendices

## A EXPERIMENT DETAILS AND SETTINGS

### A.1 DETAILED OVERALL EXPERIMENTAL SETTINGS

Table 4: Prompt templates used in this paper.

| Dataset | Prompt Template (Unit) | Label Tokens |
|---|---|---|
| SST-2 | `sentence: [input sentence] sentiment: [label token] \n` | negative, positive |
| MR | `review: [input sentence] sentiment: [label token] \n` | negative, positive |
| FP | `sentence: [input sentence] sentiment: [label token] \n` | negative, neutral, positive |
| SST-5 | `sentence: [input sentence] sentiment: [label token] \n` | poor, bad, neutral, good, great |
| TREC | `question: [input sentence] target: [label token] \n` | short, entity, description, person, location, number |
| AGNews | `news: [input sentence] topic: [label token] \n` | world, sports, business, science |

**Prompt Template.** We conduct experiments on a specific prompt template for each dataset as shown in Table 4. Moreover, similar to typical ICL practices, we reduce the label into one token to simplify the prediction decoding. The reduced label tokens are also shown in Table 4.

**Quantization.** In our experiments, we use `BitsAndBytes`[9] to quantize Llama 3 70B and Falcon 40B to `INT4`. For the other models, full-precision inference is conducted.

**Other.** All the experiment materials (models and datasets) are downloaded from `huggingface`. For the BGE M3, we use its `pooler_output` as the output feature.

### A.2 CALCULATION OF MUTUAL NEAREST-NEIGHBOR KERNEL ALIGNMENT

In this paper, we need to measure the similarity between features from two different models or model layers. There are many approaches (Klabunde et al., 2023), and we use mutual nearest-neighbor kernel alignment (Huh et al., 2024), which is relatively concise and accurate, calculated as follows to measure the similarity of representation from the same object set $\mathcal{X} = \{x_i\}_{i=1}^n$ in different feature spaces.

**Calculation of Mutual Nearest-neighbor Kernel Alignment**. Given a representation mapping $\delta : \mathcal{X} \to \mathbb{H}^d$ from the objects to a space where similarity measurement $\langle \cdot, \cdot \rangle : \mathbb{H}^d \times \mathbb{H}^d \to \mathbb{R}$ is defined, we can calculate the similarity map from dataset $\mathcal{X}$ as $\mathcal{S}_\delta \in \mathbb{R}^{n \times n}$, where the elements are $\mathcal{S}_{\delta|i,j} = \langle \delta(x_i), \delta(x_j) \rangle$, especially, we axiomatic define $\langle x, x \rangle = 1$, so we set the diagonal element $\mathcal{S}_{\delta|i,i} \doteq 0$ since they are trivial values that disturb the following calculation with values 1.

Given two representation functions $\delta_1$ and $\delta_2$, two similarity map can be calculated as $\mathcal{S}_{\delta_1}$ and $\mathcal{S}_{\delta_2}$ on the same object set $\mathcal{X}$. For each line vector index $i = 1, 2, \ldots, n$ in $\mathcal{S}_{\delta_1}$, we select **the index of top-$K$ elements from greater to lower** as $\text{top}_k(\mathcal{S}_{\delta_1|i})$. Similarly, we get $\text{top}_K(\mathcal{S}_{\delta_2|i})$ from $\mathcal{S}_{\delta_2}$.

Then, we calculate the kernel alignment for sample $i$ as:

$$\text{KA}_\mathcal{X}(\delta_1, \delta_2)_i = \frac{\left| \text{top}_K(\mathcal{S}_{\delta_1|i}) \cap \text{top}_K(\mathcal{S}_{\delta_2|i}) \right|}{k}. \tag{1}$$

The kernel alignment for dataset $\mathcal{X}$ is the average on each $\text{KA}_\mathcal{X}(\delta_1, \delta_2)_i$.

**Implementation.** In our experiments, we choose cosine similarity as the $\langle \cdot, \cdot \rangle$, and $K \doteq 64$. According to experiment settings in §2.2, $n \doteq 512$ is defined, and a randomlized matrix $\mathcal{S}$ have KA $= 64/512 = 0.125$ as the random baseline.

### A.2.1 BACKGROUND VALUES OF KERNEL ALIGNMENT: LABEL TOKEN - TEXT ENCODING.

In a specific layer of a decoder Transformer, the representations of a token $x$ can be written as $\delta(x) = e(x) + \epsilon(p)$, where $e(x)$ is the embedding vector of the token $x$, and $\epsilon(p)$ are the residual side-flow w.r.t. the context $p$. Given two specific tokens $x_i$ and $x_j$ where kernel alignment is calculated

---

[9] https://huggingface.co/docs/bitsandbytes/main/en/index

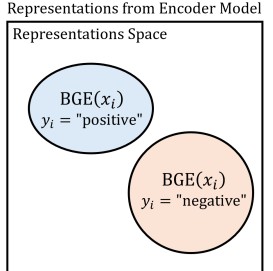 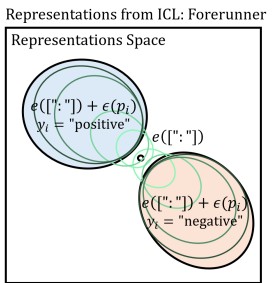 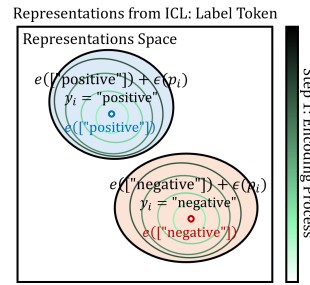

Figure 9: Distributions (*clusters*) of representations generated by: **Left**: encoder model (BGE), clustering w.r.t. the label. **Middle**: ICL on the forerunner token, where representations gather into one point ($e([": "])$) when no Transformer operation is conducted. **Right**: ICL on the label token, where representations gather into points w.r.t. label (using a 2-way example) when no Transformer operation is conducted, causing a high background value.

from different ICL-styled input sequences $p_i$ and $p_j$, we discuss the background value of kernel alignment as below.

**Intuition.** As shown in Fig. 9, the hidden states on the label token have a prior clustering, making them naturally similar to the representation generated by the encoder model, even if the ICL process does not encode it sufficiently. So, at layer 0, since the model is not able to perform any contextual encoding in this layer, the kernel alignment on the forerunner is on a random baseline value of 0.125, but the value on the label token will be greater (estimated below). In the case where (1) BGE generates fully linearly separable clusters against $x$ in each class, with sufficient inter-cluster distance, and (2) the number of samples for each label is not less than 64, such upper bias can be expected to be $0.125(|\mathbb{Y}|-1)$ (proof omitted). Intuitively, such bias can propagate along the residual network from the embedding of $[y]$ to every layer, making the results of every layer upper-biased.

**Similarity on Forerunner Tokens.** The cosine similarity between $\delta(x_i)$ and $\delta(x_j)$, where the $x_i$ and $x_j$ are forerunner tokens of the input sequence can be written as:

$$\langle \delta(x_i), \delta(x_j) \rangle = \langle e(x_i) + \epsilon(p_i), e(x_j) + \epsilon(p_j) \rangle \tag{2}$$

$$= \frac{\langle e(x_i), e(x_j) \rangle + \langle e(x_i), \epsilon(p_j) \rangle + \langle e(x_j), \epsilon(p_i) \rangle + \langle \epsilon(p_i), \epsilon(p_j) \rangle}{\|e(x_i) + \epsilon(p_i)\|_2 \|e(x_j) + \epsilon(p_j)\|_2}. \tag{3}$$

Denote $B_{i,j} = \langle e(x_i), \epsilon(p_j) \rangle + \langle e(x_j), \epsilon(p_i) \rangle$, $C_{i,j} = \|e(x_i) + \epsilon(p_i)\|_2 \|e(x_j) + \epsilon(p_j)\|_2$, and notice that $x_i = x_j$ since they are forerunner tokens which is kept consistent in experiments, we have:

$$\langle \delta(x_i), \delta(x_j) \rangle = \frac{1 + B_{i,j} + \langle \epsilon(p_i), \epsilon(p_j) \rangle}{C_{i,j}}. \tag{4}$$

**Similarity on Label Tokens.** Similarly, the cosine similarity between $\delta(y_i)$ and $\delta(y_j)$ on label tokens can be written as:

$$\langle \delta(y_i), \delta(y_j) \rangle = \frac{\langle e(y_i), e(y_j) \rangle + B_{i,j} + \langle \epsilon(p_i), \epsilon(p_j) \rangle}{C_{i,j}}. \tag{5}$$

**Encoding on Label Tokens Enhances the Similarity with Same Labels.** Given $K = 1$ for simplicity, the probability $\text{top}_1\left(\mathcal{S}_{\delta|i}\right)$ selects a sample with the similar label with $i$-th sample **on the forerunner token** can be written as:

$$\mathrm{P}_F\left[y_i = y_{\text{top}_1(\mathcal{S}_{\delta|i})}\right] = r_{y_i} \frac{\mathbb{E}_{j|y_i=y_j}\left[\langle \delta(x_i), \delta(x_j) \rangle\right]}{\mathbb{E}_j\left[\langle \delta(x_i), \delta(x_j) \rangle\right]} \tag{6}$$

$$\approx r_{y_i} \frac{1 + \mathbb{E}_{j|y_i=y_j}\left[B_{i,j}\right] + \mathbb{E}_{j|y_i=y_j}\left[\langle \epsilon(p_i), \epsilon(p_j) \rangle\right]}{1 + \mathbb{E}_j\left[B_{i,j}\right] + \mathbb{E}_j\left[\langle \epsilon(p_i), \epsilon(p_j) \rangle\right]}, \tag{7}$$

where the $r_{y_i}$ is the label ratio of $y_i$. Similarly, the probability on the label token can be written as:

$$
\begin{aligned}
&P_L \left[ y_i = y_{\text{top}_1(\mathcal{S}_{\delta|i})} \right] \\
&\approx \frac{r_{y_i}(1 + \mathbb{E}_{j|y_i=y_j}[B_{i,j}] + \mathbb{E}_{j|y_i=y_j}[\langle \epsilon(p_i), \epsilon(p_j) \rangle])}{\begin{array}{l} r_{y_i}\left(1 + \mathbb{E}_{j|y_i=y_j}[B_{i,j}] + \mathbb{E}_{j|y_i=y_j}[\langle \epsilon(p_i), \epsilon(p_j) \rangle]\right) \\ + (1-r_{y_i})\left(\langle e(y_i), e(y_j) \rangle + \mathbb{E}_j[B_{i,j}] + \mathbb{E}_j[\langle \epsilon(p_i), \epsilon(p_j) \rangle]\right) \end{array}} \\
&\geqslant P_F \left[ y_i = y_{\text{top}_1(\mathcal{S}_{\delta|i})} \right] .
\end{aligned}
\tag{8}
$$

That is, the inputs with the same labels with $x_i$ are easier to be selected into the $\text{top}_1(\mathcal{S}_{\delta|i})$. Notice that we make approximations here: (1) we consider the $\mathbb{E}_{j|y_i=y_j}[C_{i,j}] \approx \mathbb{E}_j[C_{i,j}]$, i.e., the 2-norm of two encoding vectors are considered equal granted by normalization used in Transformer. (2) We consider the context term $c$ in the label token scenario the same as the forerunner scenario since the difference is only a label token, which usually occupies quite a small part of the input sequence.

**Background Values of Kernel Alignment.** According to the explanation above, as shown in Fig. 9, it is intuitive to conclude that $\text{top}_K(\mathcal{S}_{\delta|i})$ from label tokens is easier to cluster samples with the same label as $y_i$. Moreover, a well-pre-trained encoder can catch the prior distribution of the input texts determined by their labels, and also cluster samples with the same label, causing a high similarity of similarity map, so that a high but unfaithful kernel alignment as the background value from the similarity on $e(y)$ but not $\epsilon(p)$. An intuitive verification of such background value is shown in the Fig. 2 (Left), where the "Label Token" curve has a high value in layer 0 with $\epsilon(p) = 0$. However, the background value also indicates that the representation generated by BGE correctly clusters the samples, confirming its reliability.

## A.3 TRAINING AND INFERENCE OF CENTROID CLASSIFIER

In this paper, we follow Cho et al. (2024) to train centroid classifiers as a probe toward hidden states of LMs. In detail, given the LM's hidden states set $\{h_i^l\}_{i=1}^m$ of the selected tokens (according to the experimental setting, the last label token or forerunner token) in layer $l$ from an [ICL input]-[query label] set $\mathcal{Z} = \{(p_i, y_i)\}_{i=1}^m$, where the labels are limited in label space $\mathbb{Y}$, in the **training phase**, we calculate the centroid of the hidden state $\bar{h}_y^l$ for each label respectively:

$$
\bar{h}_y^l = \mathbb{E}_{i|y_i=y}[h_i^l] .
\tag{9}
$$

In the **inference phase**, we extract the equitant[10] hidden state $h_t^l$ as the training phase from the test input, and calculate the similarity between $h_t^l$ and the centroids calculated above. Then, we choose the label of the most similar centroids as the prediction:

$$
\mathcal{C}(h_t^l) = \underset{y}{\arg\max} \langle h_t^l, \bar{h}_y^l \rangle .
\tag{10}
$$

**Implementation.** In our experiments, we set training sample number $m \doteq 256$, similarity function $\langle a, b \rangle \doteq -\|a - b\|_2$.

## A.4 CARTOGRAPHY DETAILS OF FIG. 6 (MIDDLE)

In Fig. 6 (Middle), we define a Correct Label Assignment, here we introduce how this measurement is calculated. Suppose we have an attention score $\mathcal{A}_{W,f}(K, Q)$ calculated as (also written as $\alpha_{K \to Q}$ in brief):

$$
\mathcal{A}_{W,f}(K, Q) = f(Q^\top W K) ,
\tag{11}
$$

with hidden dimensionality of $d$, give a certain layer, $K \in \mathbb{R}^{d \times n_t}$ is the hidden state matrix of full context, $Q \in \mathbb{R}^{d \times 1}$ is the hidden state of query's forerunner token ($Q^\top = K_{n_t}^\top$), $f : \mathbb{R}^{n_t} \to \Omega^{n_t}$ is a normalization mapping from $n_t$-dimensional real vector to $n_t$-dimensional probability vector (usually softmax function), the $W$ is a linear kernel, usually $W_Q^{h\top} W_K^h$ for multi-head attention or $I = \text{diag}(\mathbb{1}^{n_t})$ for vanilla attention.

---

[10]Equitant refers to hidden states from the same layer and token type. While, in experiments shown in Fig. 5 (Right), we don't keep the token type consistent.

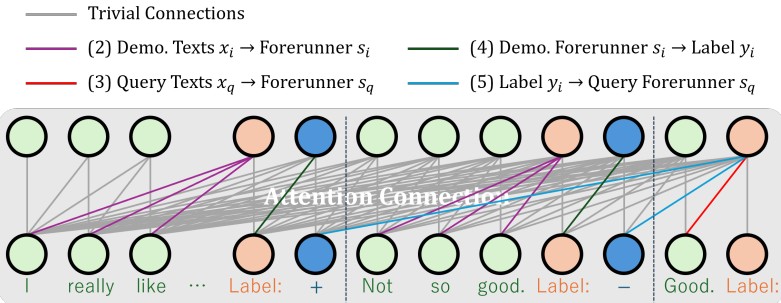

Figure 10: Visualization of non-trivial attention connection defined and disconnected in ablation analysis in §5.1 and Table 1. Notations and line numbers are same as Table 1.

For one input sample, given the token-index set of label tokens as $\mathcal{L}$, the token-index set of label tokens which is the same as the query's ground truth label as $\mathcal{L}^+$, we define the **C**orrect **L**abel **A**ssignment (CLA) of one sample as:

$$\text{CLA}_{W,f}(K,Q,\mathcal{L},\mathcal{L}^+) = \frac{\sum_{i \in \mathcal{L}^+} \mathcal{A}_{W,f}(K,Q)_i}{\sum_{i \in \mathcal{L}} \mathcal{A}_{W,f}(K,Q)_i}. \tag{12}$$

Intuitively, CLA reflects the accuracy of attention computation $\mathcal{A}_{W,f}$ towards label tokens on one input. For an input set built from a dataset, we calculate the averaged CLA on these inputs, and repeat in every layer to plot a curve of Averaged CLA against layer numbers. Specifically:

**(1) Vanilla attention.** We assign $W \doteq I$, $f$ to linear normlization.

**(2) Best Induction Head.** For each attention head $h$, we assign $W^h \doteq W_Q^{h\top} W_K^h$, $f \doteq \text{softmax}$. For each input, we calculate $\max_h \text{CLA}_{W^h,f}(K,Q,\mathcal{L},\mathcal{L}^+)$ as the result for single input.

**(3) Head Average.** For each attention head $h$, we assign $W^h \doteq W_Q^{h\top} W_K^h$, $f \doteq \text{softmax}$. For each input, we calculate $\sum_h \text{CLA}_{W^h,f}(K,Q,\mathcal{L},\mathcal{L}^+)/|\mathcal{H}|$, where the $|\mathcal{H}|$ is the amount of heads in current layer, as the result for single input.

Note that we do not consider the absolute value of attention assignment on label tokens in this experiment, and most of the heads have little scores assigned to the label (Fig. 6 (Left)), therefore, although the average assignments tend to be average, this result shown in Fig. 6 (Middle) does not contradict the phenomenon that ICL can achieve high accuracy.

## A.5 CARTOGRAPHY DETAILS OF FIG. 7

For Fig. 7, we input one sample from SST-2 into Llama 3 70B, take the output of layer 30 on the label tokens to span a matrix $K_{\mathcal{L}}$, and map them by $W_Q^{h\top} W_K^h$ of head 32 (the best induction heads in this layer) and 9 (the worst induction heads) of layer 31 (the layer with the most correct induction heads), respectively. We visualize the distribution of these mapped $W_Q^{h\top} W_K^h K_{\mathcal{L}}$, we conduct principal component analysis on them, and plot them on the plane of the first two components.

For each point $q \in \mathbb{R}^2$ on the principal component plane, we calculate the attention assignment as follows. Give the index set of "positive" label token in $K_{\mathcal{L}}$ as $\mathcal{L}^+$, the index set of "negative" label token as $\mathcal{L}^-$, we calculate the attention assignment, which can be an estimate of ICL prediction (Wang et al., 2023), as:

$$\text{AttAssign}(q) = \sum_{i \in \mathcal{L}^+} q^\top W_Q^{h\top} W_K^h K_{\mathcal{L}|i} - \sum_{i \in \mathcal{L}^-} q^\top W_Q^{h\top} W_K^h K_{\mathcal{L}|i}. \tag{13}$$

We map this value to the degree of blue color of each pixel. The larger the positive value, the bluer it is, and the smaller the negative value, the redder it is.

## A.6 CARTOGRAPHY DETAILS OF FIG. 8

For Fig. 8, we calculate the magnitude of Step 1 as the finite differences of kernel alignment in Fig. 2 (Left, "Forerunner Token of Label"). We directly use the head counting of Fig. 5 (Middle)

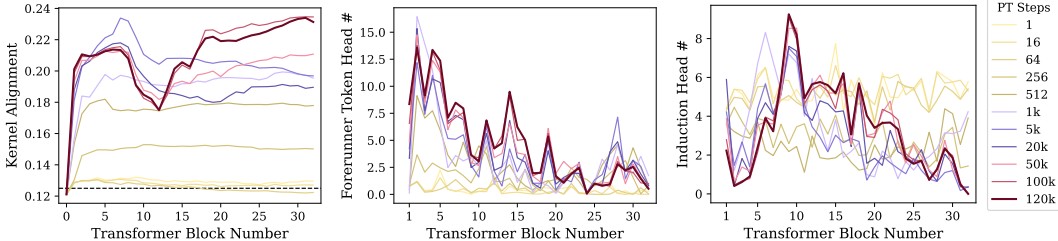

Figure 12: The 3 operating magnitudes on Pythia 6.9B with various pre-training steps. **Left**: Step 1, Input Text Encode. **Middle**: Step 2, Semantics Merge. **Right**: Step 3, Feature Retrieval and Copy, measured by the correct induction head numbers.

and Fig. 6 (Left) as the magnitude of Steps 2 and 3. These data are regularized and converted into transparencies.

### A.7 EXPERIMENT SETTING OF ABLATION EXPERIMENTS IN TABLE 1

In Table 1, we attribute each step of the inference process to specific attention connections, also shown in Fig. 10. When we aim to remove this step from the inference, we eliminate (i.e. zeroing) all corresponding attention connections from layer 0 to layer $\{25\%, 50\%, 75\%, 100\%\} \times \text{TotalLayers}$.

For reference, for each experiment, we also conduct controlled experiments where the same amount of randomly selected attention connections are removed from the same layers as the controlled values. The controlled results are shown as smaller numbers under the experimental results.

## B  LM PRE-TRAINING DYNAMICS MEASURED BY ICL CIRCUIT

We extend the discussion of pre-training dynamics in §6 here.

One can divide a self-regression model into an early part and a later part in depth of layers, where the early part encodes the input into a hidden representation, and the later part decodes the hidden representation back to the input. So, the training object can also be divided into an encoding loss and a decoding loss. According to the discussion in §6, the operation of Step 1 can be classified as encoding, and the other two steps of the induction circuit can be classified as decoding. Intuitively, since the decoding operations require the encoding results as input, unless the encoding operation converges to a stable output, the decoding can not be trained since the input-output mapping is noised, causing unstable gradients to interfere with the training (Liu et al., 2020).

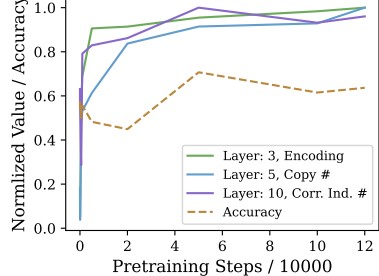

Figure 11: Operating magnitude for each inference step (normalized) and ICL accuracy against pre-training steps on Pythia 6.9B and SST-2.

We confirm such a hypothesis by measurements in Pythia 6.9B (Biderman et al., 2023) as shown in Fig 11, where: (1) The magnitude of the 3 operations emerge in the early phase of pre-training (less than 10k steps), and is monotonically increasing, while the encoding operation has the fastest growth rate. (2) ICL capacity appearance after all three operations reaches a high level (around 50k steps, notice that the random accuracy is 0.5), while the curve morphologies of the operating magnitudes against the layer numbers (shown in Fig. 12) are convergence to the last training step. Such results suggest that: LMs start to produce the inner encoding in the very early steps of the pre-training, and can be an important fundamental in building the subsequent induction circuits, as explained in the previous works mentioned in §2.1.

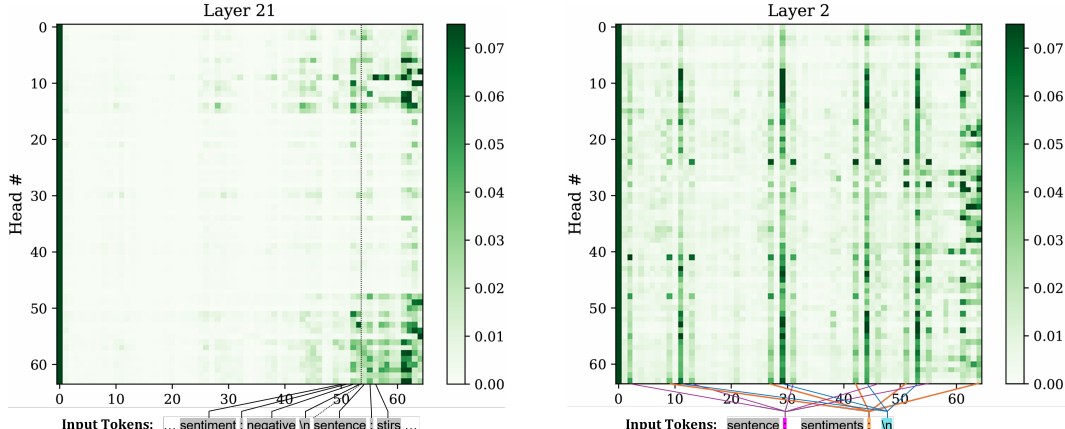

Figure 13: Attention score visualized with the forerunner token of the ICL query as the attention query, from (**Left**) layer 21 and (**Right**) layer 2 in Llama 3 70B from an input case.

## B.1 DATA DISTRIBUTION REQUIREMENT EXPLAINED BY HIDDEN STATE MULTIPLEX

Moreover, the hidden space multiplexing in the induction operation observed in this paper can give a prototypical and phenomenological conjecture for the data distribution requirement found in the previous works (Olsson et al., 2022; Reddy, 2024; Singh et al., 2024b), where data with a large label space and various tasks can promote ICL and suppress **In-w**eight **L**earning (IWL), and vice versa. Intuitively, suppose the encoding inputted into the later layers is clustered by their labels (similar to the well-embedded input styles in the works above, confirmed in Fig. 3). In that case, we can say a cluster center is the **eigen-subspace of the corresponding label**. Since attention only conducts dot-multiplication operations, let us assume that these eigen-subspaces are radially distributed.

During the training, (1) **When the label space is small**, the trained attention heads only need to extract the projected length of the query on each label's eigen-subspace. For each label, such operation has a parameters' analytical solution with (*encoding kernel*) $W_Q^\top W_K = \mathrm{I}$ and (*decoding transformation*) $W_O W_V = o_y^\top e_y$, where $o_y$ is the label token's output embedding, and $e_y$ is the label's eigen-subspace. From such an operation, theoretically, one layer can handle at most $|\mathcal{H}|$ labels, where $|\mathcal{H}|$ is the head amounts. While, considering the sparsity of these eigen-subspaces, such an upper bound can be increased to $d'|\mathcal{H}|$ by multiplex one head to decode an orthogonal group of labels with orthogonal eigen-subspaces of $\mathcal{E} = [e_{y_1}; e_{y_2}; \ldots; e_{y_{d'}}]$ and orthogonal output embedding $\mathcal{O} = [o_{y_1}; o_{y_2}; \ldots; o_{y_{d'}}]$, where $d'$ is the inner dimension of attention head, with decoding transformation $W_O W_V = \mathcal{O}^\top \mathcal{E}$. (2) **When the label space expands**[11], the decoding transformation can not distinguish all the clusters since the $W_O W_V$ is low-rank. Driven by the training loss, the model can choose to transform the encoding kernel to focus on catching the most similar label tokens with the query, and copy the label tokens' information back to the query. As a result, one attention head can catch at most $d'$ groups of label tokens mapped collinearly by the encoding kernel (note that this set of labels may not appear simultaneously in the context, so confusion can be avoided), and the common space of these label words become the induction subspace shown in §4.2.

Our other conjecture is that the ICL training endpoint is thermodynamically stable (with a lower loss), and in contrast, the IWL training endpoint is kinetically stable (with a more accessible training trajectory). Moreover, the IWL training object can be a precursor of ICL training, since the total number of labels fed into the model gradually increases with the data. So, we can hypothesize that: When the metastable state is disturbed by some condition, such as the appearance of rare or noisy labels, the training can show a phase transition toward a thermodynamically stable state.

---

[11]Notice that such a situation can also occur when the orthogonality between eigen-subspaces or output embeddings is lost. A common situation is that the variance of the cluster increases, creating confusion within the decoding space of the attention head.

Notice that this section is our hypothesis based on the results of this paper, the detailed dynamics are still unknown, which should be empirically validated in future work. One can start by decomposing pre-training targets into implicit tasks, and examine how these tasks can evoke the occurrence of the 3-step inference operations. A possible beginning is: finding implicit input-label tuples in wild data.

## C   CAN LMS SEGMENT ICL INPUTS?

The experiments in §3.1 imply that LMs conduct effective segmentation on ICL-styled inputs, enabling LMs to block the interference of preceding demonstrations in the input text encoding operation. Here, as a prototypical discussion, we confirm the existence of the segmentation operation and then reveal that such segmentation can be done in very early layers from an attention operation focusing on some specific segmentation tokens in the inputs.

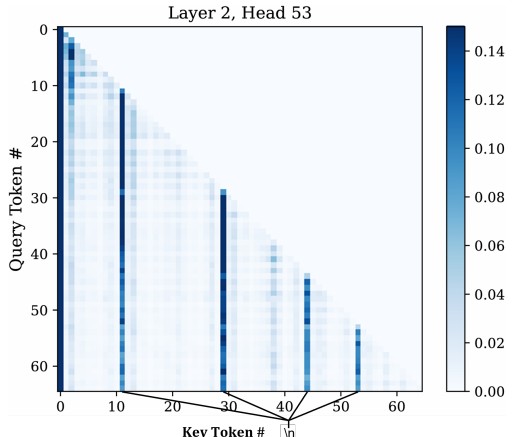

Figure 14: Attention matrix visualized on layer 2, head 53 in Llama 3 70B from an input case.

As a preliminary observation, we visualize the attention scores with the last forerunner token as the attention queue at layer 21 (a layer with high encoding magnitude, refer to Fig. 2) from an input case, as shown in Fig. 13 (Left), where most of the attention heads are focused on the query text. The visualization suggests that encoding operations are localized into the query tokens.

Although position embedding inserts sufficient positional information to hidden states and enables attention heads to identify nearby tokens, we believe that position embedding is insufficient to accurately segment the various input parts of uncertain lengths. We hypothesize that LM focused on natural delimiters (e.g. "\n", ": ") in the input during the early stages of inference, and visualization in Fig. 13 (Right) supports such hypothesis: In layer 2, most of the attention heads focus on the natural delimiters, and another visualization in Fig. 14 shows that all attention queries (not only the forerunner token) exhibit similar separator-focusing behavior, suggesting that: Some attention heads merge all the preceding delimiters' representation into every token as *delimiter-based positional encoding*, making the representations of tokens with the same number of preceding delimiters similar, while differentiating the representations of tokens with different numbers of preceding delimiters. In the subsequent inference process, LM can utilize these delimiter-based positional encodings for localization operations. Such observation is also consistent with Fig. 4.

Furthermore, we empirically demonstrate that delimiters have significant saliency towards ICL accuracies in Table 5 (upper), experimented by removing them from prompt templates. Interestingly, the trial to completely remove these delimiters from the inputs yielded almost random results, even though these inputs still conform to the primary form of ICL. A reliable reason can be that: The hiatus of the delimiter interferes with the encoding operation (Step 1) on both demonstrations and queries, which completely disrupts the ICL process.

Table 5: Accuracy drop with delimiters removed/modified from prompts on SST-2.

| Template Modification | Acc. (%) |
|---|---|
| **None** (Table 4) | 91.60 |
|  - w/o "\n" | 93.36 |
|  - w/o ": " | 85.74 |
|  - w/o "sentence: ", "sentiment: " | 79.10 |
|  - w/o all above | 50.98 |
|  - ": " → "hello " | 78.91 |
|  - ": " → "@ " | 91.41 |
|  - ": " → "positive " | 71.48 |
| **(Random)** | 50.00 |

The scale of such a segmentation operation can surprise one, since more than half of the heads focus on the segmenting operation as shown in Fig. 13 (Right). However, as an assumption, we want to argue that dividing the input text into local segmentation is a crucial step in language modeling, so, functionally, LM has sufficient "motivation" to focus on segmenting by the pre-training objective. Moreover, based on the above principles, as long as the delimiter appears periodically at appropriate positions and can be captured by attention heads (only in the structured parts of the prompt template), as shown in Table 5 (lower), the delimiter can be designed to any token. While we still recommend natural delimiters without semantics in the template design.

## D    SEMANTICS MERGE IS NON-SELECTIVE ON LOCATION

To investigate whether the copy processing described in §4.1 has selectivity on the forerunner token, on every attention head of each layer, for a given token position $i$, we measure the **N**ormalized **C**opy **M**agnitude shown below:

$$\text{NCM}_{n_t}(\alpha) = n_t \alpha_{(i-1)\rightarrow i}, \tag{14}$$

where the $\alpha_{(i-1)\rightarrow i}$ is the attention score with $i$-th token serving as the attention query and $(i-1)$-th token serving as the attention key. For each layer, we export the NCM at all positions and on all attention heads, and separately statistics the cases where the $i$-th token is a label token or a non-label token. The results for 4 models on SST-2 are shown in Fig. 15.

From the results, no significant statistical differences between these two types of tokens can be observed, suggesting that the semantics copy process, which is identified as Step 2 of our circuit, is not selective on the token types. However, even if we demonstrate that the model cannot exhibit selectivity in the copy positions within the current input, a potential direction for future research is to investigate whether any forerunner tokens in the inputs can enhance or weaken this copy process. Given that this copying is known to be related to label denoising (§4.1), exploring and carefully designing these forerunner tokens could significantly benefit the control of ICL behavior.

## E    MEASUREMENT ON DIRECT DECODING

This section measures the direct decoding bypass, suggesting that: (1) Direct decoding on well-processed hidden states with some later layer skipped can get satisfactory accuracy even better than the full inference process. (2) Direct decoding on insufficient processed hidden states adds bias towards the predicting distribution.

We examine the first claim by applying the language model head on each layer's hidden state on SST-2, Llama 3 8B, and conduct a standard ICL process on the decoded token prediction distribution. The results are shown in Fig. 16, where direct decoding accuracy emerges from random to near 1 around layer 18. Refer to Fig. 8 and results in Appendix H.2, we can confirm: Accuracy emerges after all three steps are executed. Moreover, the accuracies on the intermediate hidden states are even higher than the last hidden states, which is aligned with the discussion in Table 3. So, we can conclude: Direct decoding on well-processed hidden states can classify well.

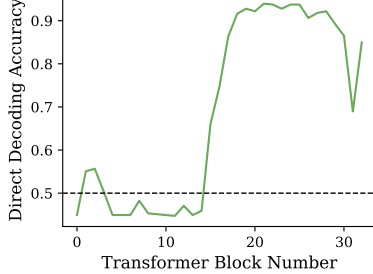

Figure 16: Direct decoding accuracies on various layers.

Moreover, we infer that direct decoding from lower layers, where hidden states are not sufficiently processed, causes prediction bias. We investigate the influence of the direct decoding result of layer 0, by the relationship between direct decoded distribution and final prediction distribution. In detail, on SST-2 and Llama 3 8B, we use various forerunner tokens with different direct decoding distributions on the label tokens "positive" and "negative", and calculate their ICL prediction probability distributions respectively, as shown in Fig. 17 (Upper), where forerunner tokens with biased direct decoding distribution produce prediction biases with the same tendencies. While, when we apply contextual calibration (Zhao et al., 2021), which removes the background value without a query from the prediction, such similar tendencies disappear (Fig. 17 (Lower)).

## F    DEMONSTRATIONS ENHANCE THE INFERENCE OF PERPLEXED QUERIES

We investigate the correlation between the queries' perplexities and the classification accuracies with and without demonstrations, as a supplement of results in Fig. 2 (Right). We divide the queries into 10 bins w.r.t. the language modeling loss, and calculate the prediction accuracy in each bin, shown in Fig. 18. In these results, although a unified correlation can not be observed, we can confirm that: Compared to the 0-shot results, the 4-shot inference shows better accuracies, especially on queries with high language modeling loss. So, we can conclude that: Demonstrations enhance the inference accuracy of perplexed queries, consistent with the results in Fig. 2 (Right).

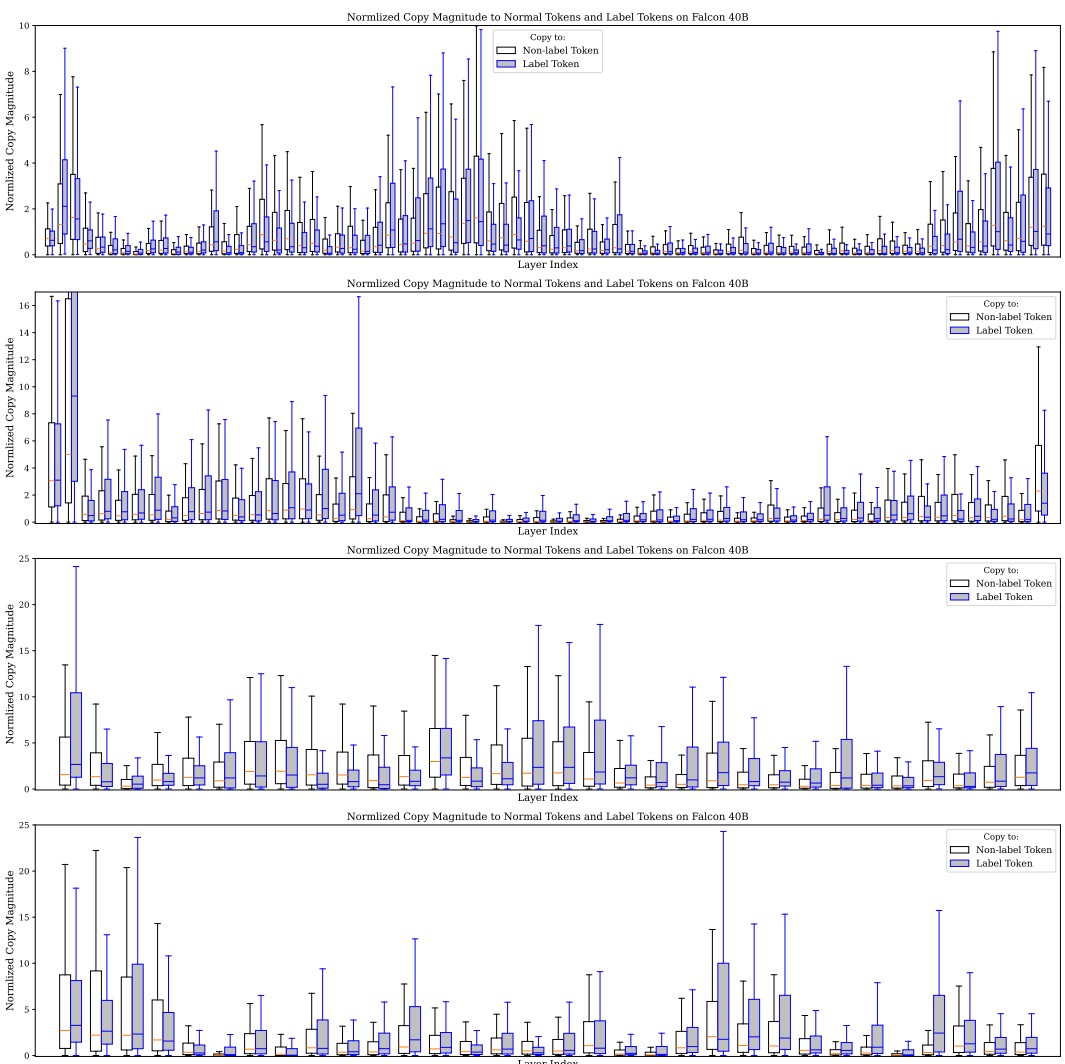

Figure 15: Copy magnitude normalized by the sequence length from the previous token to the non-label token and label token, for every token and head in each layer. Significant statistical differences cannot be observed.

# G  DEGRADATION ANALYSIS: IN-WEIGHT LEARNING WITHOUT GROUND-TRUTH LABEL IN CONTEXT

Given the circuit proposed in this paper, it is intuitive that some label tokens, especially the ground-truth label token of the query should be presented in the demonstrations, which is the typical in-context learning setting compared to the **I**n-**W**eight **L**earning (IWL) setting (Chan et al., 2022; Reddy, 2024) where the ground-truth label is not offered in the demonstration. When the ground-truth label is missing, the behavior of Steps 1 and 2 remains constant as they are independent of the query (due to the causal attention mask), while the behavior of Step 3 should be re-discussed.

In this section, we demonstrate that our circuit, especially the induction processing of Step 3, can still explain the inference behavior in the IWL scenario, but is not as robust as in the ICL scenario. Moreover, our experiments demonstrate that in the IWL scenario, induction heads are detrimental to the model's predictions, which is consistent with the conclusion drawn from the results in the main text. Also, some inference phenomena in the IWL scenario can be explained by our conclusions. While, we also illustrate through a counterexample that under the condition of IWL, even if the induction heads work negatively, the demonstrations can still enhance the inference.

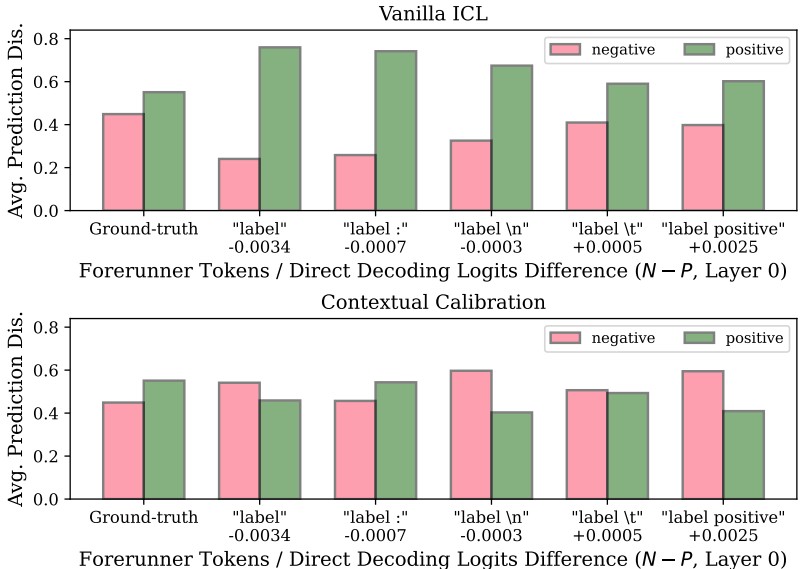

Figure 17: Predicting distributions on different forerunner tokens with various direct decoding logits. Inference process used: **Upper**: Vanilla ICL, **Lower**: Biased removed inference by Contextual Calibration proposed by Zhao et al. (2021).

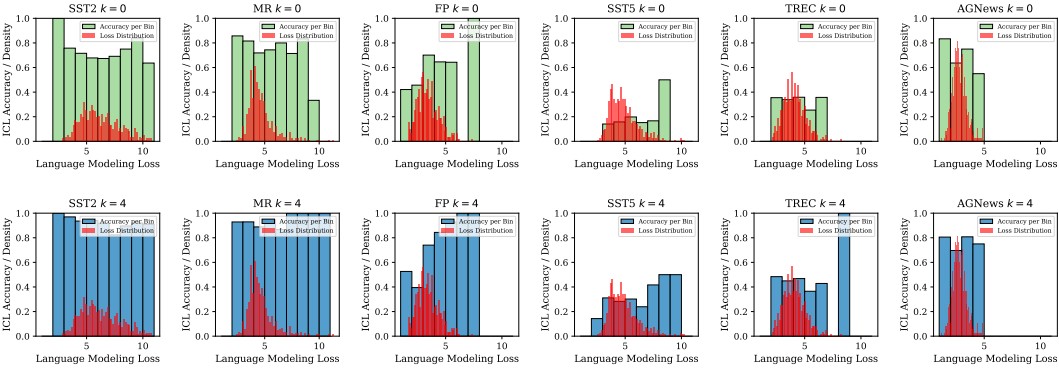

Figure 18: The correlations between language modeling loss and ICL prediction accuracies. **Upper**: 0-shot results; **Lower**: 4-shot results.

Experimental results in this section require us to consider new inference dynamics for IWL scenarios, although we believe the discussion for ICL scenarios in this paper is robust. We demonstrate that such separate consideration of ICL and IWL is reasonable and responsible by highlighting the essential differences between ICL and IWL settings, as also shown in previous works (Chan et al., 2022; Reddy, 2024).

## G.1  HOW ACCURATE CAN INDUCTION HEADS EXPLAIN IWL INFERENCE?

Our main concern is: Whether the induction heads in our circuit for ICL can explain or predict the inference behavior in the IWL scenario. We design a metric to investigate the accuracy of such an explanation: We calculate the divergence between the real output probability of the model and the predicted output probability through the attention assignment of the induction heads. In detail, given a label space $\mathbb{Y}$, for each label $l \in \mathbb{Y}$, we denote the token-index set of $l$ as $\mathcal{L}_l$. Then, we calculate the predicted output probability $\hat{o}$ as:

$$\hat{o} = \text{softmax}\left(\left[o_1, o_2, \ldots, o_{|\mathbb{Y}|}\right]\right), \tag{15}$$

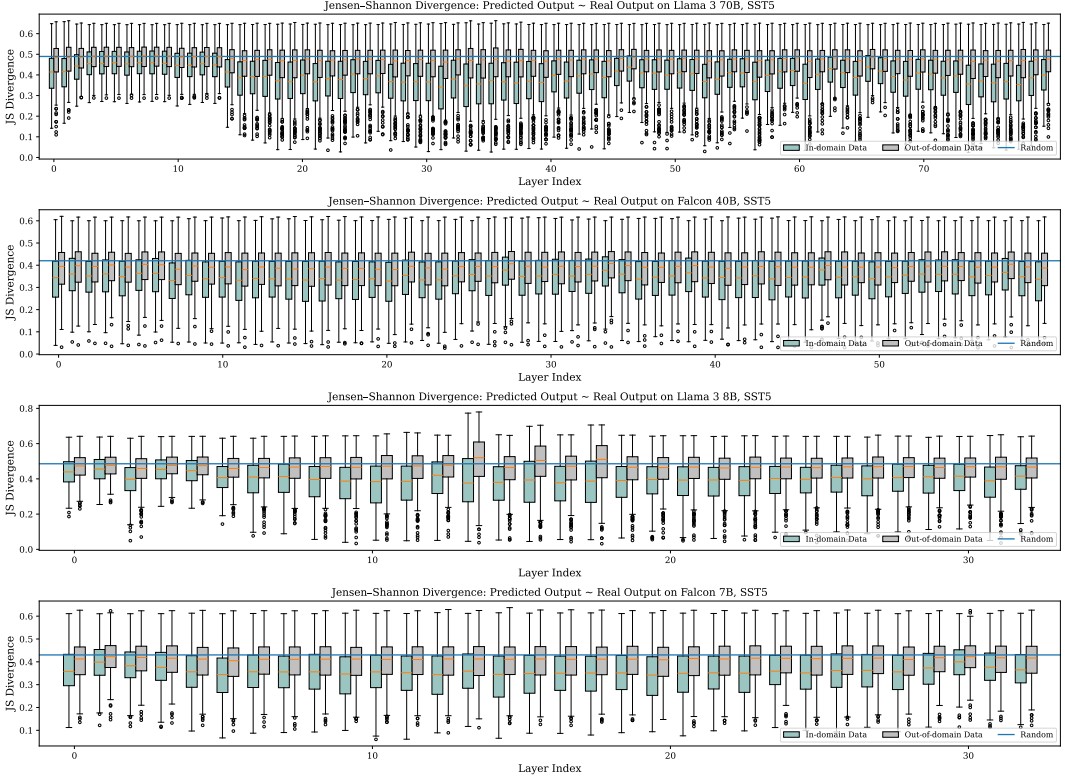

Figure 19: The JS divergence between the predicted output by the induction attention score (Eq. 15) and real output on ICL and IWL settings. Since not all attention heads are induction heads, the lowest 5 data of each layer is selected.

$$o_l = n_t \sum_{i \in \mathcal{L}_l} \alpha_{i \to q}^h, \tag{16}$$

where the $\alpha_{i \to q}^h$ is the attention score of head $h$ where the query's forerunner token serves as the attention query, and the $i$-th token as the attention key. Given the real output distribution of ICL as $o$, we calculate the Jensen–Shannon divergence $D_{\mathrm{JS}}[\hat{o}, o]$ as the metric for the accuracy of predicting $o$ from $\hat{o}$. For each Transformer layer, we select the results of the 5 attention heads with the lowest divergence, and the results on SST-5[12] are shown in Fig. 19.

In the results, statistically, compared to the ICL setting, induction heads provide poorer explanations of model predictions in the IWL setting. However, these divergences consistently remain below the random baseline, indicating that model outputs in the IWL setting can still be weakly explained by induction heads. This indicates that the circuit utilized in this paper aligns with the IWL scenario, albeit slightly weaker.

### G.2 DO INDUCTION HEADS CONTRIBUTE TO IWL PROCESSING?

To test the generalizability of our conclusion towards IWL scenarios, we conduct more ablation experiments similar to §5.1, but with filtered inputs with no ground-truth query label presented in the context. The results are shown in Table 6. Generally speaking, these experiments confirm that: Our theoretical framework is still able to explain inference phenomena in IWL scenarios, thus strengthening our framework.

In detail, compared to the results of unfiltered inputs (Table 1), we highlight 4 major phenomena, and explain them within our framework: (1) The baseline accuracies (line 1) in IWL settings are lower than in normal settings, even worse than random prediction. This is easy to understand given

---

[12]We select a more-way task to reduce the influence of frequency bias of ICL (introduced in §5.3).

Table 6: Accuracy variation (%) with each inference step ablated on **Left**: Llama 3 8B, **Right**: Falcon 7B **in IWL scenarios**. Notations are the same with Table 1, significant comparisons are highlighted with arrows.

**Left: Llama 3 8B**

| # | Attention Disconnected Key → Query | Affected Layers Ratio (from layer 1) | | | |
|---|---|---|---|---|---|
| | | 25% | 50% | 75% | 100% |
| 1 | **None** (4-shot IWL baseline) | ±0 (Acc. 33.43 ↓) | | | |
| | *– Step1: Input Text Encode –* | | | | |
| 2 | **Demo. Texts** $x_i$ → **Forerunner** $s_i$ | −2.31 (+0.03) | −17.90 (+0.78) | −28.58 (+0.49) | −31.22 (−0.10) |
| 3 | **Query Texts** $x_q$ → **Forerunner** $s_q$ | −2.67 (+0.23) | −12.96 (+0.23) | −17.87 (−0.46) | −13.41 (+0.26) |
| | *– Step2: Semantics Merge –* | | | | |
| 4 | **Label** $y_i$ → **Query Forerunner** $s_q$ | −1.20 (+0.29) | +0.13↑ (−0.10) | +0.52↑ (−0.06) | +0.88↑ (−0.06) |
| | *– Step3: Feature Retrieval & Copy –* | | | | |
| 5 | **Label** $y_i$ → **Query Forerunner** $s_q$ | +2.08↑ (−0.23) | +2.05↑ (−0.06) | +5.40↑ (−0.36) | +22.95↑ (+0.00) |
| | *Reference Value* | | | | |
| 6 | **Zero-shot** | +17.22 (Acc. 50.65) | | | |
| 7 | **Random Prediction** | −0.93 (Acc. 32.50) | | | |

**Right: Falcon 7B**

| # | Attention Disconnected Key → Query | Affected Layers Ratio (from layer 1) | | | |
|---|---|---|---|---|---|
| | | 25% | 50% | 75% | 100% |
| 1 | **None** (4-shot IWL baseline) | ±0 (Acc. 27.41 ↓) | | | |
| | *– Step1: Input Text Encode –* | | | | |
| 2 | **Demo. Texts** $x_i$ → **Forerunner** $s_i$ | −17.64 (−0.13) | −24.32 (−0.42) | −26.17 (−1.24) | −26.43 (+0.91) |
| 3 | **Query Texts** $x_q$ → **Forerunner** $s_q$ | −3.19 (−0.33) | −15.33 (−0.23) | −21.48 (+0.03) | −22.53 (+0.32) |
| | *– Step2: Semantics Merge –* | | | | |
| 4 | **Demo. Forerunner** $s_i$ → **Label** $y_i$ | +2.93↑ (−0.00) | +1.10↑ (+0.32) | +6.38↑ (0.13) | +5.86↑ (−0.13) |
| | *– Step3: Feature Retrieval & Copy –* | | | | |
| 5 | **Label** $y_i$ → **Query Forerunner** $s_q$ | +8.20↑ (+0.13) | +5.63↑ (0.23) | +9.28↑ (+0.00) | +29.26↑ (+0.10) |
| | *Reference Value* | | | | |
| 6 | **Zero-shot** | +33.58 (Acc. 60.99) | | | |
| 7 | **Random Prediction** | +5.09 (Acc. 32.50) | | | |

the frequency bias introduced in §5.3. (2) As we gradually suppress the induction heads (line 5), the accuracies increase, which is consistent with the conclusion in the main text of this paper: In IWL setting, induction heads can not find and copy any correct label-related information in the context, and only copy the label information presented in the context noisily. Therefore, after removing it from the inference process, the accuracies increase and approach a 0-shot level. These results confirm our main claim of this section: Induction heads are detrimental to IWL inference. (3) Interestingly, removing the copying processing (line 4) unexpectedly enhances the accuracy. We infer that the input texts of different labels under the same task still have a certain background-value similarity of the encoding at the forerunner token. Such similarities are propagated into the label tokens by Step 2, enhancing the attention flow from labels to the query's forerunner token in Step 3, causing a decrease in accuracy. (4) Removing the demonstration text encoding (line 2) decreases the accuracy to around 0. According to our discussion about Step 1, in this case, forerunner tokens cannot receive information from the demonstration text, leading an equivalent sequence for the later layers without demonstration text like $[s_1][y_1][s_2][y_2]\ldots[s_k][y_k][x_q][s_q][y_q]$. Given that the forerunner token $[s_\cdot]$ is consistent throughout the input, later layers are likely to conduct induction only on the forerunner token, whose subsequent tokens do not contain the correct label, leading to a 0 accuracy. This is also a critical rebuttal to previous work (Wang et al., 2023), which believes the label tokens directly collect the information of input text: If that is the case, then disconnecting the links from demonstration texts to the forerunner tokens will not have such a significant disturbance on the accuracy.

In summary, these results indicate: Even in the IWL setting, this paper's conclusions can explain the inference phenomena, and the findings in our paper are also significantly enhanced. This extends the applicability of our conclusions, while, given the existence of some counterexamples shown below, we still hope that future work can provide a more detailed discussion of the inference dynamics under the IWL scenario.

## G.3 THE COUNTEREXAMPLE

Given the demonstration shown in Fig. 20, we use the query "Geoffrey Hinton", whose ground-truth label is presented in the context as "Researcher" as the ICL example, and the query "Michael Jordan", whose ground-truth label "Athlete" is NOT presented in the context as an IWL example. Here, to investigate the behavior of the induction heads, we input both examples to Llama 3 70B, and visualize the attention scores from the query's forerunner token (which serves as the attention query) on layer 31, which is identified as the layer with the highest induction magnitude, as shown in Fig. 20. In the right part of the figure for the IWL scenario, the attention magnitude directed toward the labeled tokens is significantly weak, with most of the attention scores being absorbed by the Attention Sink (Xiao et al., 2024) of the first token. A comparison with the left part, where a ground-truth label is given in the context can particularly highlight such an observation. Such an observation is currently aligned with our expectations since no label features similar to the query's forerunner token can be accessed in the IWL input.

In other words, the induction heads are almost not writing demonstration-relevant information to the query in the IWL scenario. It is intuitive to infer that the model cannot predict the label for "Michael

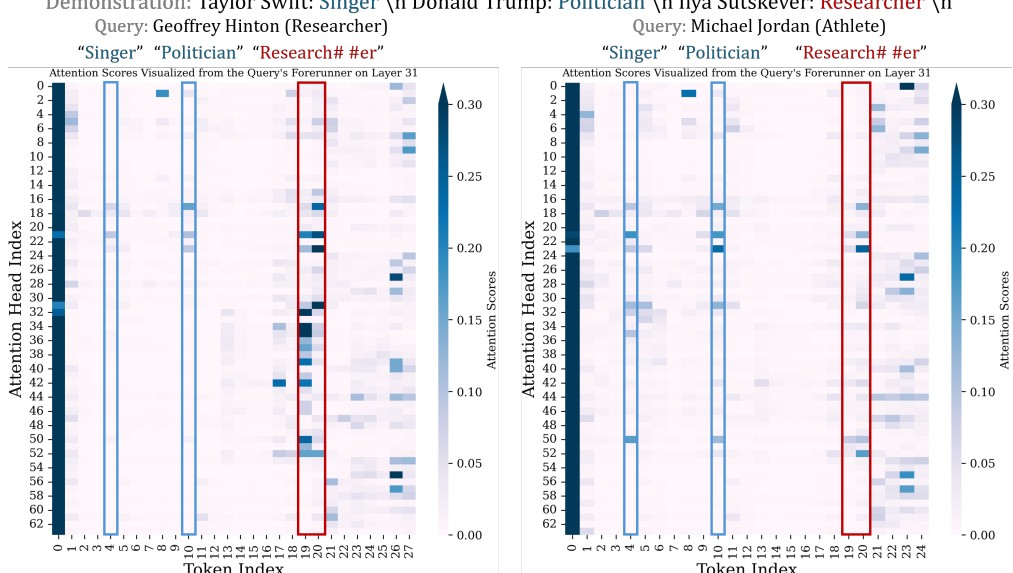

Figure 20: A counterexample when the induction head's behavior cannot predict the LM's inference behavior. Given the demonstration shown in the figure, the attention scores from the label's forerunner token are visualized on layer 31 of Llama 3 70B. The **left** part is a standard ICL scenario where the ground-truth label of the query can be accessed in the demonstrations. The **right** part is the IWL scenario where the ground-truth label of the query is not presented in the demonstrations. A clear induction pattern can not be observed in the IWL scenario.

Jordan" well, and the demonstrations cannot help the prediction either. However, as shown in Table 7, the model produces good predictions and benefits from the demonstration, which contradicts our expectations. Such contradiction indicates that even if our inference circuit can robustly explain the inference behavior in the ICL scenario, it can not generalize to the IWL scenario generalizatively.

### G.4 DISCUSSIONS: THE POSSIBILITY OF DIFFERENT INFERENCE DYNAMICS IN IWL SCENARIOS

Our explanation for this counterexample is: It can be considered that in the IWL settings, some different inference dynamics contribute to the prediction. Such difference is natural, since even if the ICL and IWL input data share a consistent format, they are fundamentally distinct, and sometimes even antagonistic, as shown in previous works (Chan et al., 2022; Reddy, 2024), which find that toy Transformers are difficult to perform well on both types of data with a same set of parameters. While, large models may allow for the coexistence of multiple inference dynamics, as discussed in §5.2, making LLMs able to yield better performance on both ICL and IWL inference scenarios.

Table 7: Label probabilities from the model predictions of the ICL (query: "Geoffrey Hinton" (Researcher)) and IWL (query: "Michael Jordan" (Athlete)) scenario shown in Fig. 20. In this case, IWL predictions also benefit from demonstrations.

| | Label Token | " Ath#" | " Research#" | " Singer" | " Politician" |
|---|---|---|---|---|---|
| IWL | 3-shot | **1.00** | 0.00 | 0.00 | 0.00 |
| | 0-shot | **0.89** | 0.04 | 0.01 | 0.06 |
| ICL | 3-shot | 0.00 | **1.00** | 0.00 | 0.00 |

Therefore, although we claim that our circuit can explain inference behavior under some of the IWL scenarios, it is reasonable to consider ICL data and IWL data separately, and this paper conducts a robust analysis under ICL conditions, leaving the IWL scenario for future works.

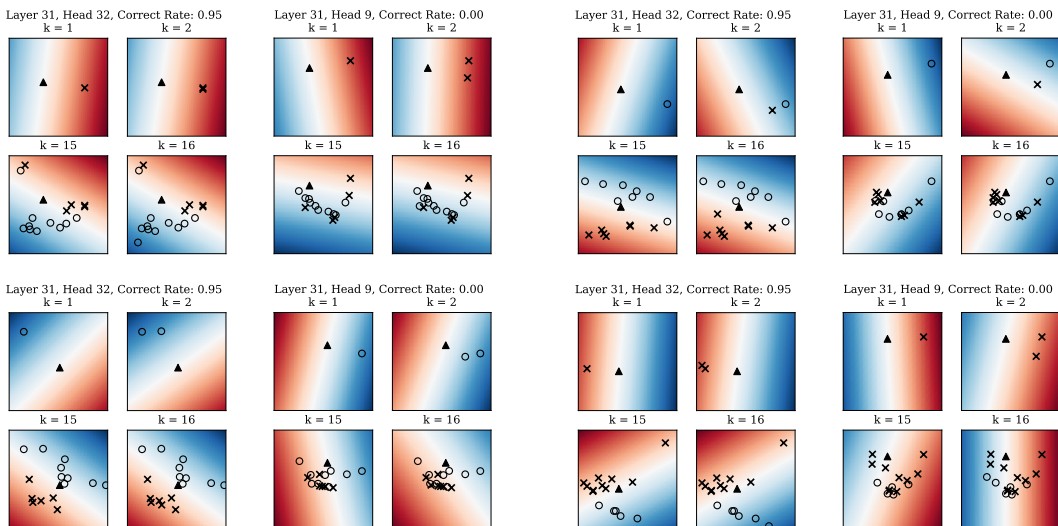

Figure 21: Supplemental experiment result on more samples for Fig. 7.

## H AUGMENTATED EXPERIMENT RESULTS

### H.1 ATTENTION HEAD STATISTICS

We count the marked times of each attention head as Forerunner Token Head (Fig. 22, 23, 24, 25) / Correct Induction Head (Fig. 26, 27, 28, 29) by all data samples from each dataset on each model.

**Forerunner Token Head Statistics.** We plot the distributions of the marked Forerunner Token Heads towards correct and wrong labels: There are observable morphological differences in figures across different datasets, while the forerunner token heads marked on the correct and incorrect labels of the same dataset are almost identical. The detailed data confirms our conclusion in §4.1.

**Induction Head Statistics.** We plot the distributions of the marked Correct Induction Heads, where there are observable morphological differences in figures across different datasets, but also significant overlaps, which confirms our conclusion in §4.2.

### H.2 OTHER LMS' EXPERIMENT RESULTS

The results of most experiments in the main text on Llama 3 8B are shown in Fig. 30, 33, 36, and 39; The results of most experiments in the main text on Falcon 40B are shown in Fig. 31, 34, 37, and 40; The results of most experiments in the main text on Falcon 7B are shown in Fig. 32, 35, 38, 41, and Table 8.

From these results, we can conclude consistently with the main text. However, as discussed in §5.2, inference dynamics on these models are delocalized, thus clear serialization of the 3 steps can not be observed in these results.

Table 8: Results of Table 1 on Falcon 7B.

| # | Attention Disconnected Key → Query | Affected Layers Ratio (from layer 1) | | | |
|---|---|---|---|---|---|
| | | 25% | 50% | 75% | 100% |
| 1 | **None** (4-shot baseline) | ±0 (Acc. 65.27) | | | |
| | *– Step1: Input Text Encode –* | | | | |
| 2 | **Demo. Texts** $x_i$ → **Forerunner** $s_i$ | $-7.65$ $-0.68 \pm 0.07$ | $-15.69$ $-0.62 \pm 0.07$ | $-27.15$ $+0.08 \pm 0.03$ | $-29.10$ $-0.36 \pm 0.07$ |
| 3 | **Query Texts** $x_q$ → **Forerunner** $s_q$ | $-8.30$ $-0.16 \pm 0.00$ | $-21.13$ $-0.15 \pm 0.00$ | $-28.84$ $+0.11 \pm 0.01$ | $-31.74$ $-0.15 \pm 0.02$ |
| | *– Step2: Semantics Merge –* | | | | |
| 4 | **Demo. Forerunner** $s_i$ → **Label** $y_i$ | $-1.01$ $+0.36 \pm 0.10$ | $-1.92$ $-0.00 \pm 0.00$ | $-1.04$ $+0.06 \pm 0.00$ | $-1.27$ $+0.06 \pm 0.02$ |
| | *– Step3: Feature Retrieval & Copy –* | | | | |
| 5 | **Label** $y_i$ → **Query Forerunner** $s_q$ | $+3.32$ $+1.72 \pm 3.94$ | $-3.61$ $-0.03 \pm 0.00$ | $-7.91$ $-0.00 \pm 0.00$ | $-5.92$ $+0.10 \pm 0.00$ |
| | *Reference Value* | | | | |
| 6 | **Zero-shot** | $-4.28$ (Acc. 60.99) | | | |
| 7 | **Random Prediction** | $-32.77$ (Acc. 32.50) | | | |

### H.3 MORE RESULTS OF FIG. 7

To enhance the persuasiveness, we additionally and randomly try 4 input samples as supplements to Fig. 7 on SST-2 and Llama 3 70B as shown in Fig. 21. From these results, we can observe similar phenomena to Fig. 21 and conclude consistently.

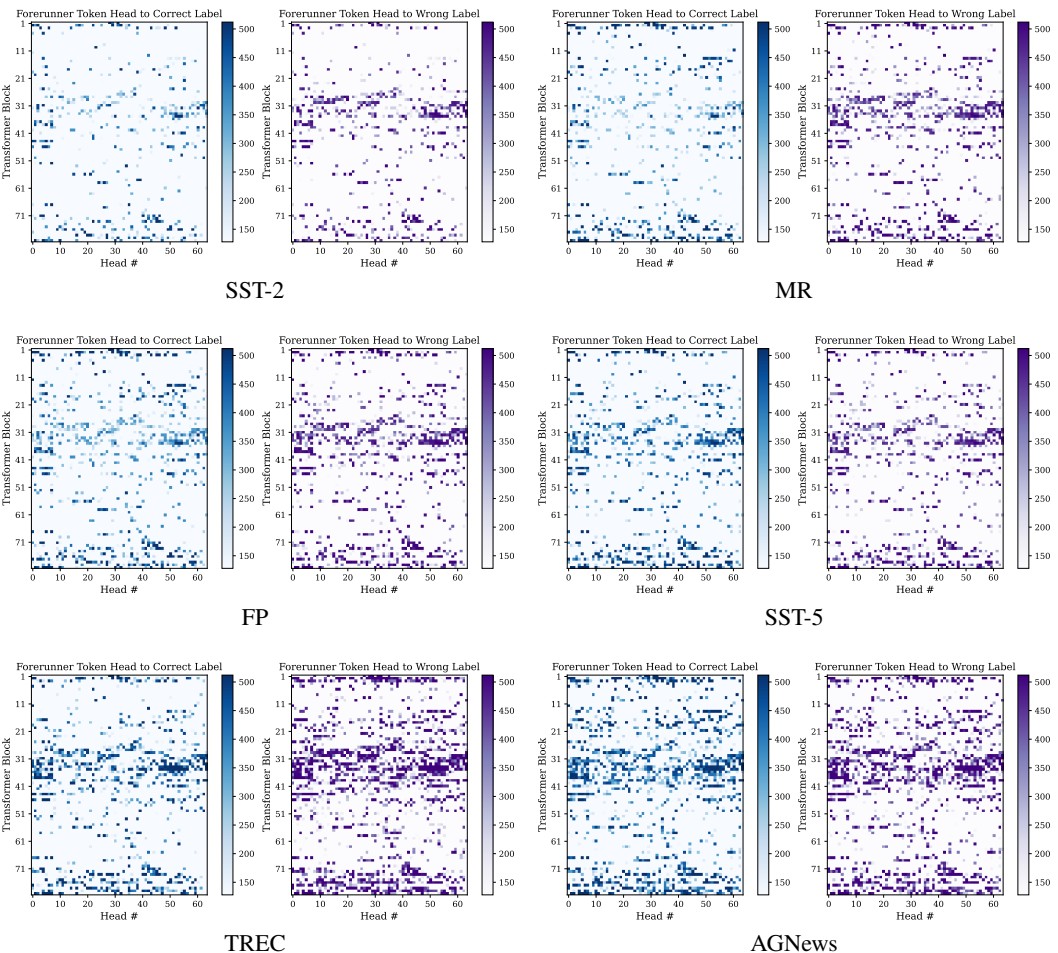

Figure 22: Forerunner Token Head marked on Llama 3 70B.

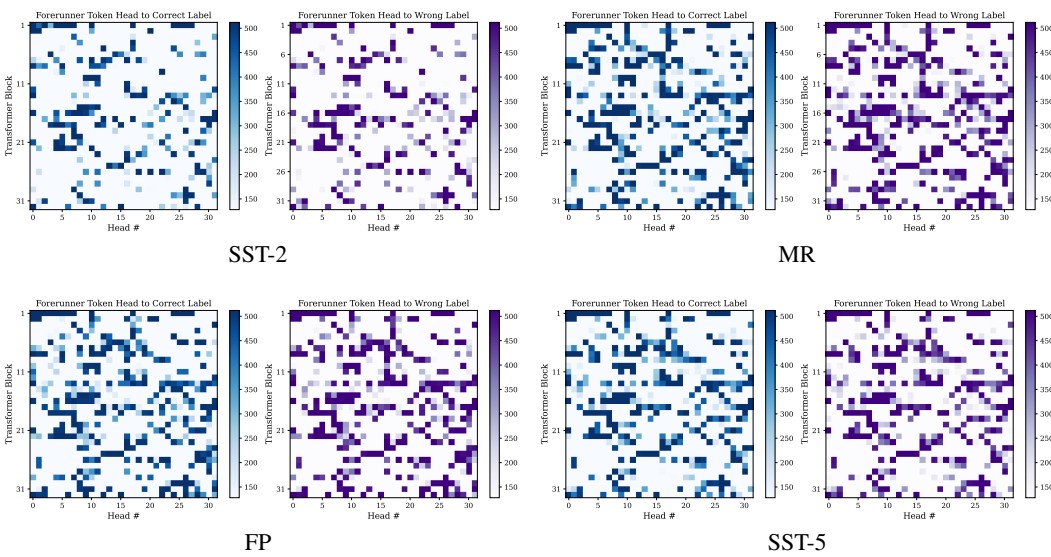

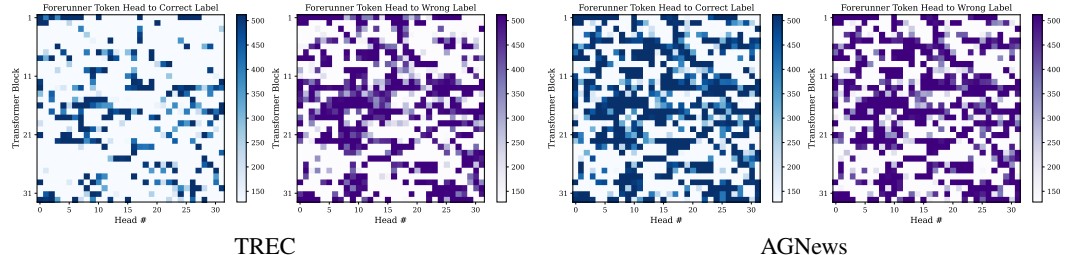

Figure 23: Forerunner Token Head marked on Llama 3 8B.

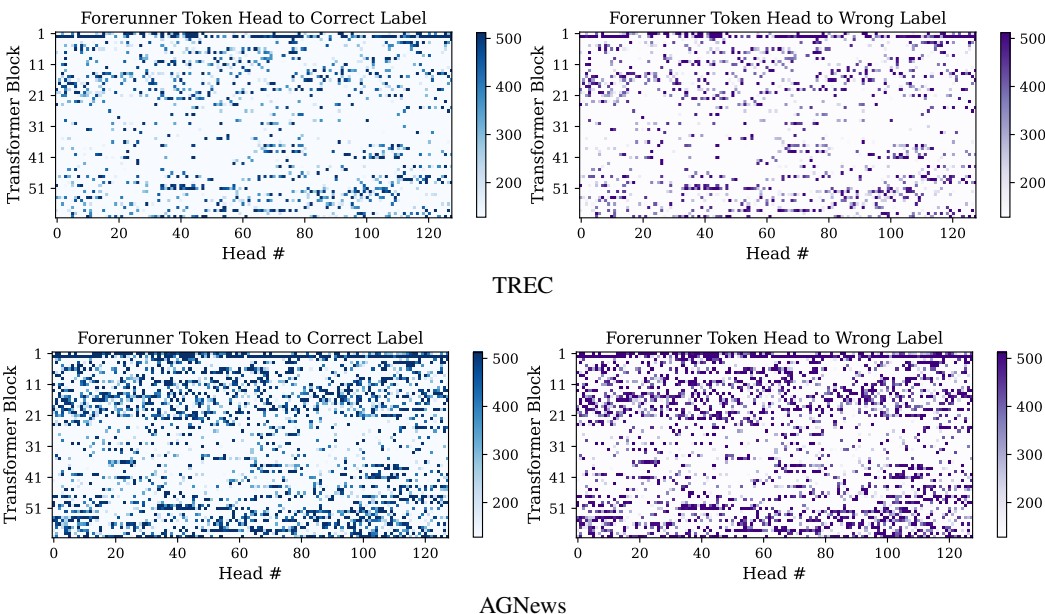

Figure 24: Forerunner Token Head marked on Falcon 40B.

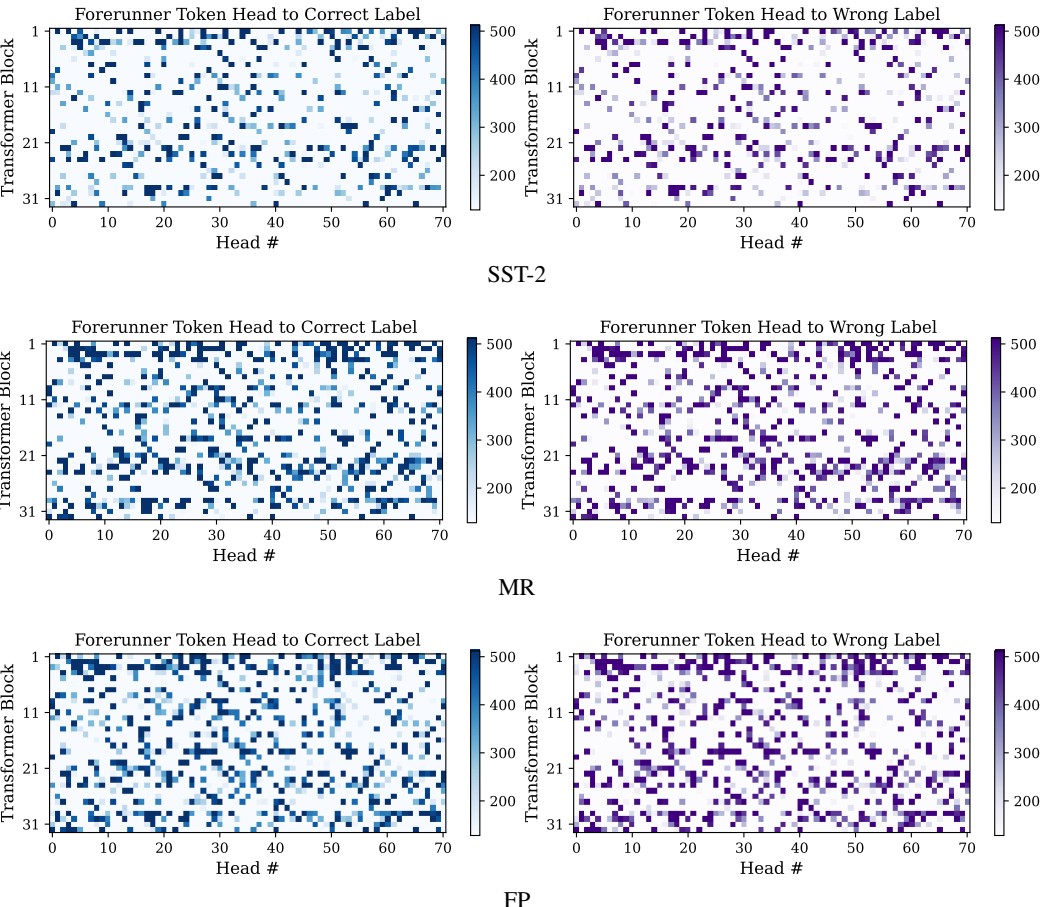

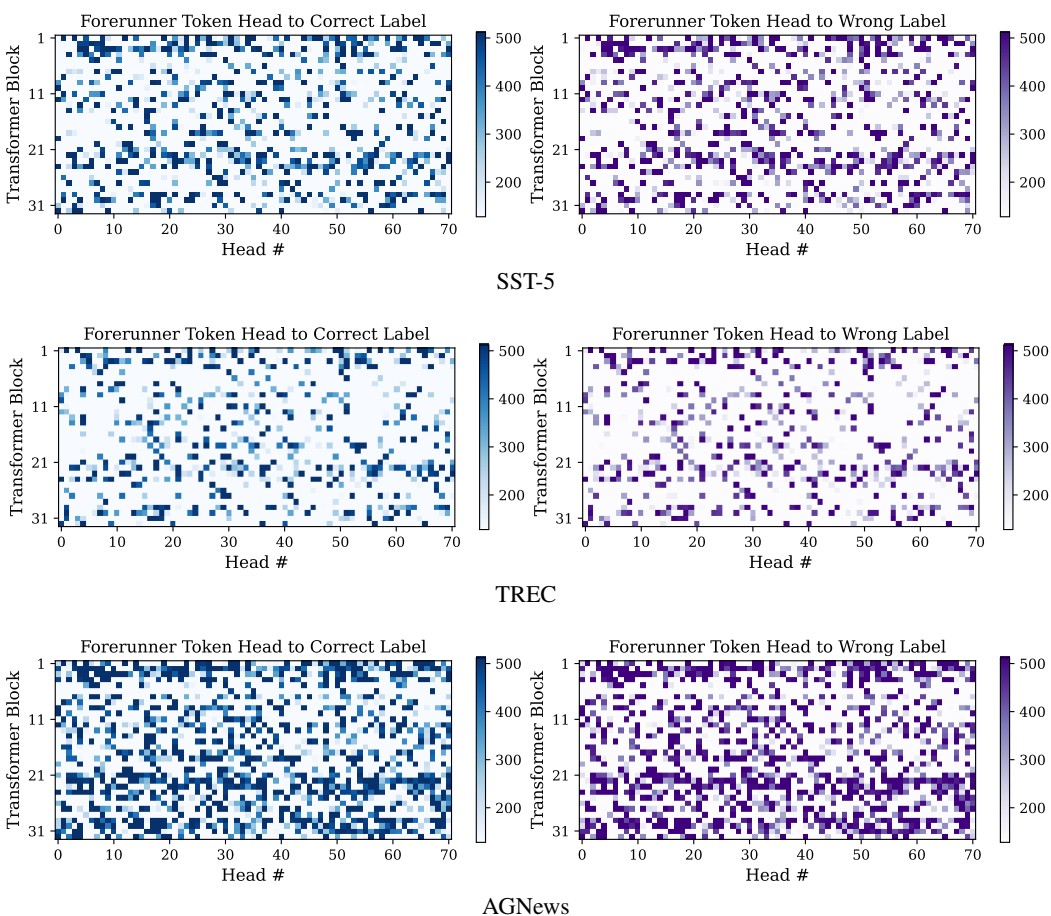

Figure 25: Forerunner Token Head marked on Falcon 7B.

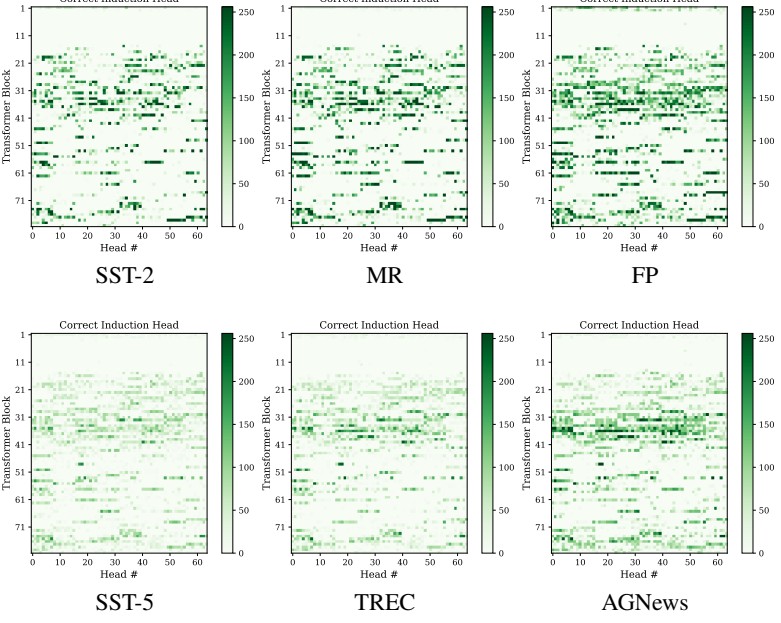

Figure 26: Correct Induction Head marked on Llama 3 70B.

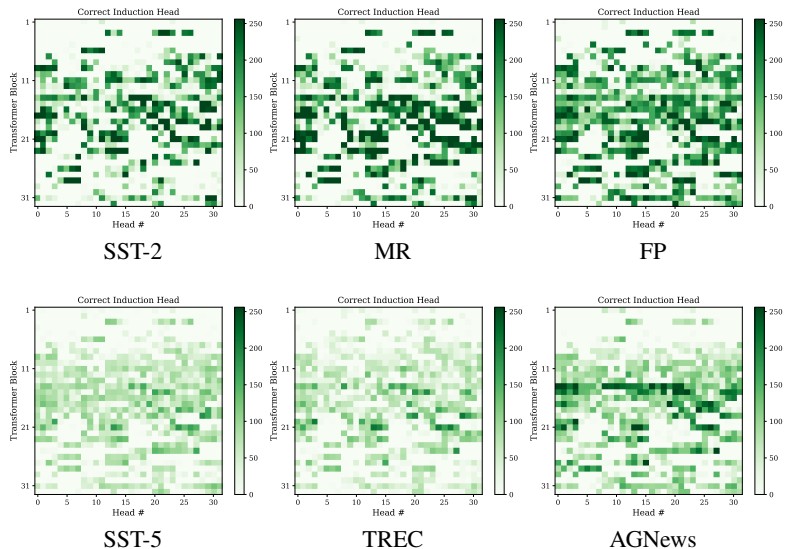

Figure 27: Correct Induction Head marked on Llama 3 8B.

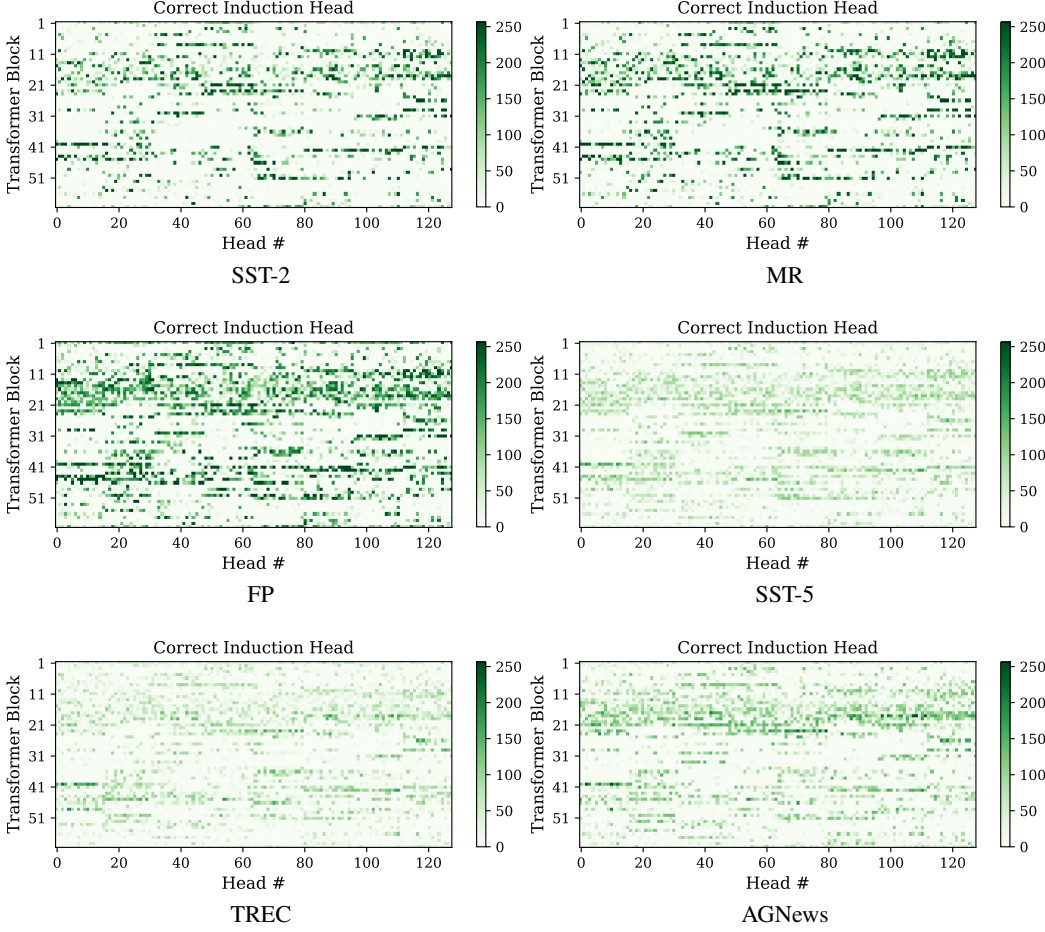

Figure 28: Correct Induction Head marked on Falcon 40B.

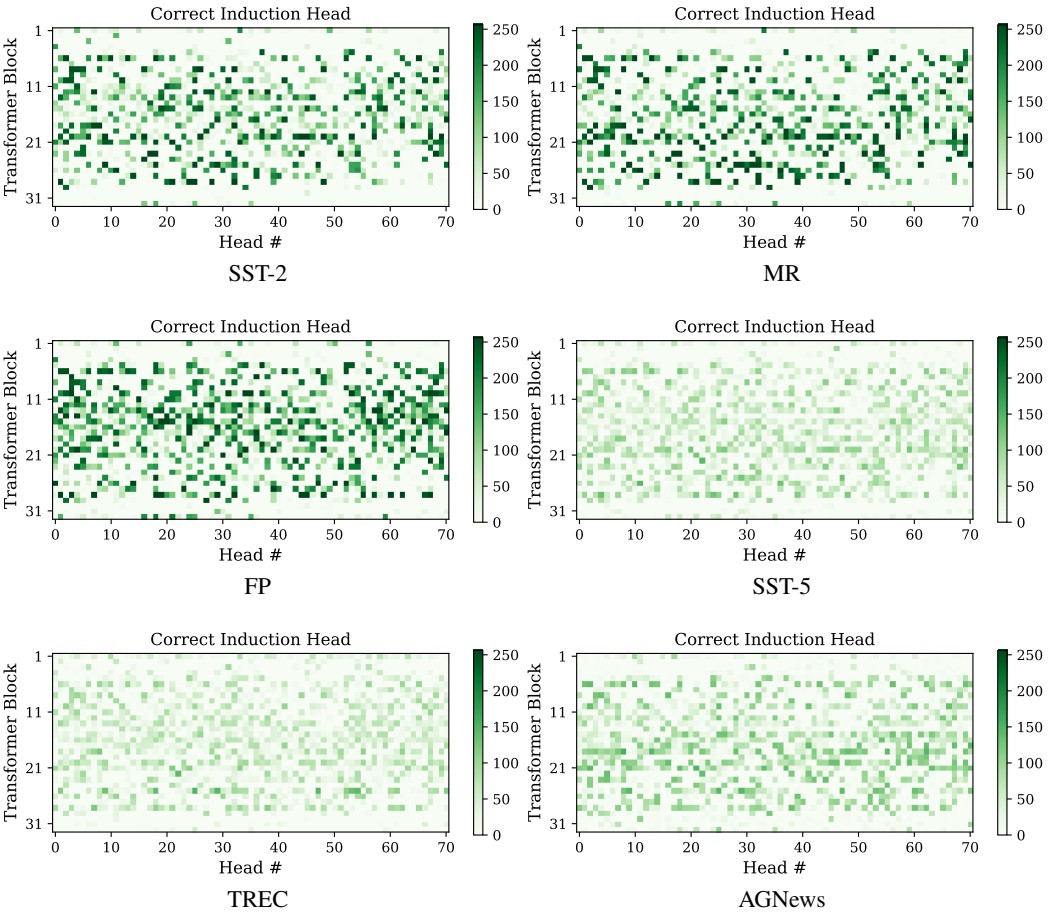

Figure 29: Correct Induction Head marked on Falcon 7B.

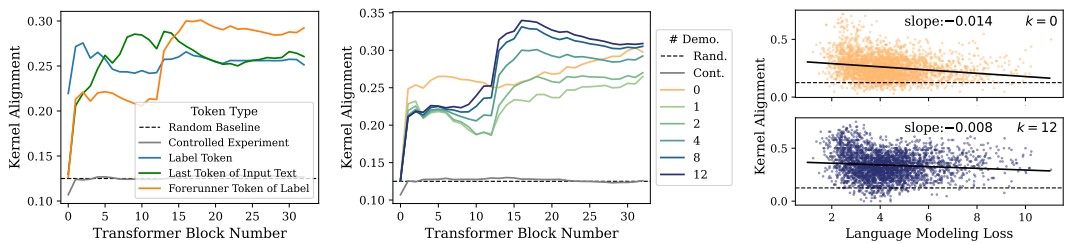

Figure 30: Augmentated results towards Fig. 2 on Llama 3 8B.

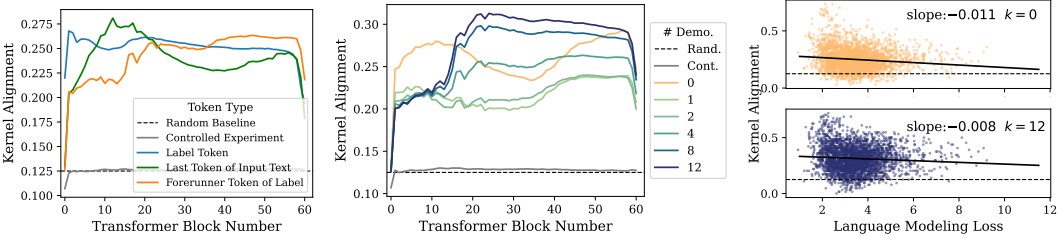

Figure 31: Augmentated results towards Fig. 2 on Falcon 40B.

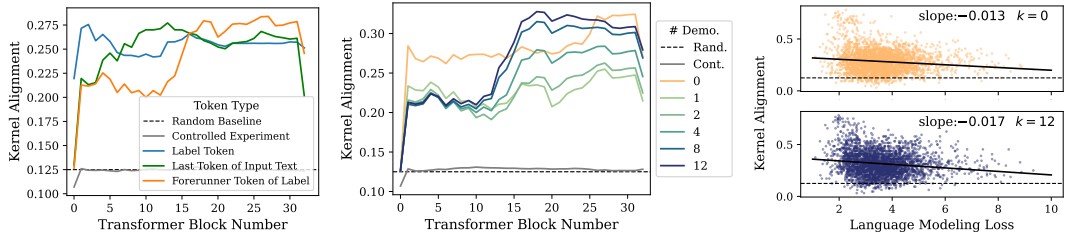

Figure 32: Augmentated results towards Fig. 2 on Falcon 7B.

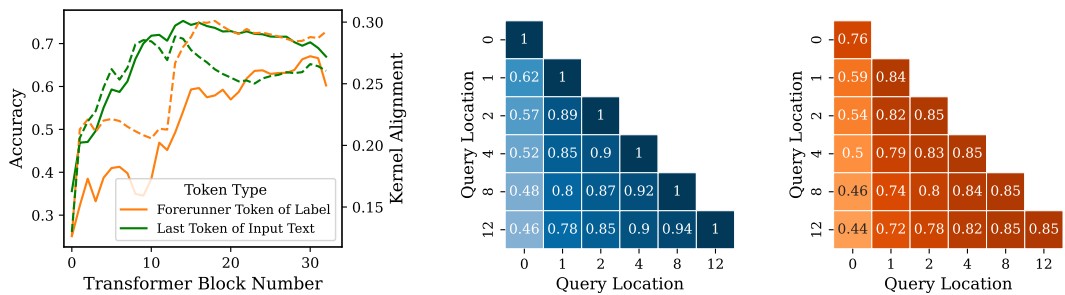

Figure 33: Augmentated results towards Fig. 3 and 4 (on Layer 16) on Llama 3 8B.

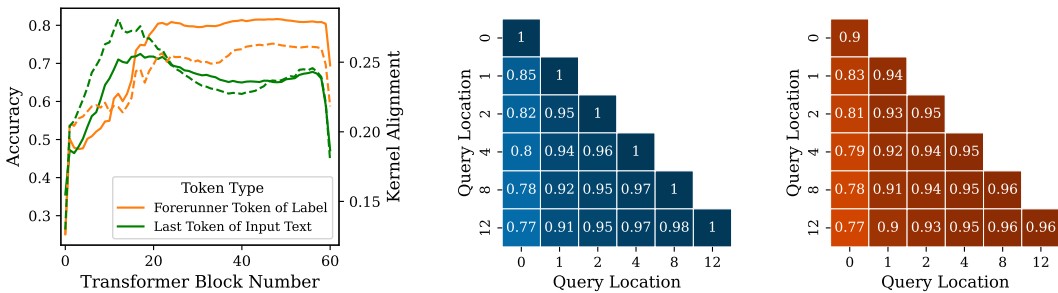

Figure 34: Augmentated results towards Fig. 3 and 4 (on Layer 24) on Falcon 40B.

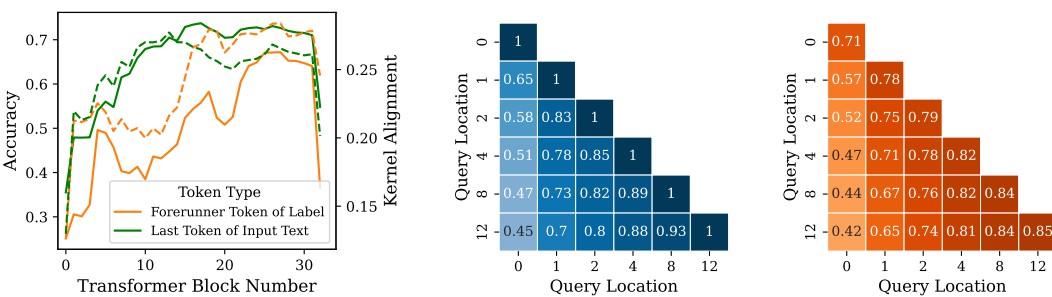

Figure 35: Augmentated results towards Fig. 3 and 4 (on Layer 16) on Falcon 7B.

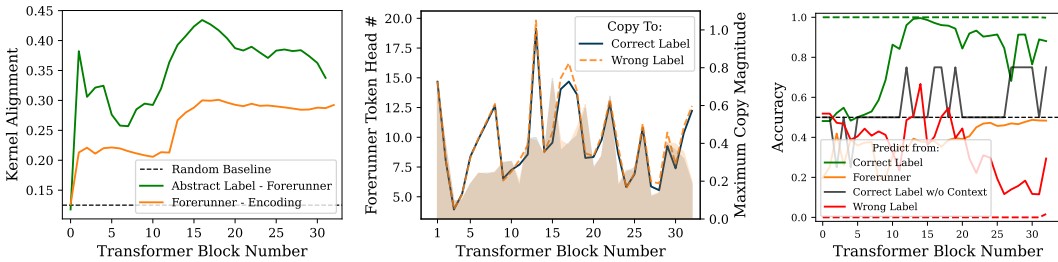

Figure 36: Augmentated results towards Fig. 5 on Llama 3 8B.

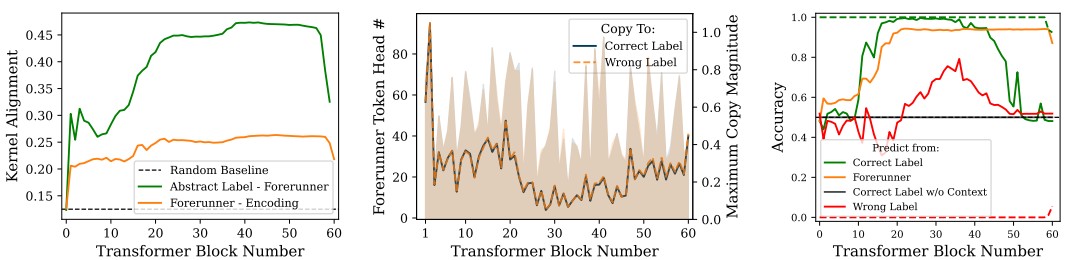

Figure 37: Augmentated results towards Fig. 5 on Falcon 40B.

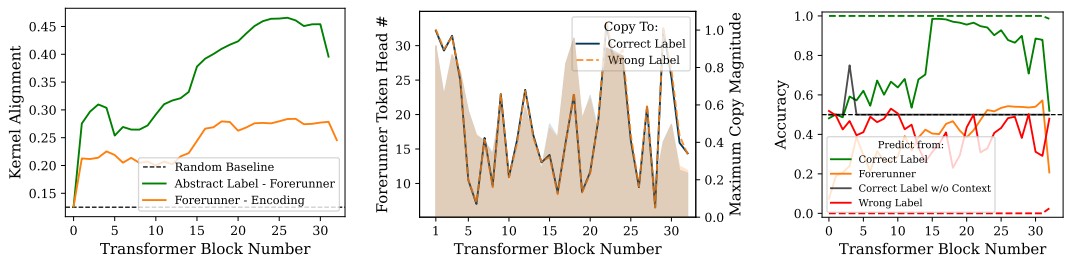

Figure 38: Augmentated results towards Fig. 5 on Falcon 7B.

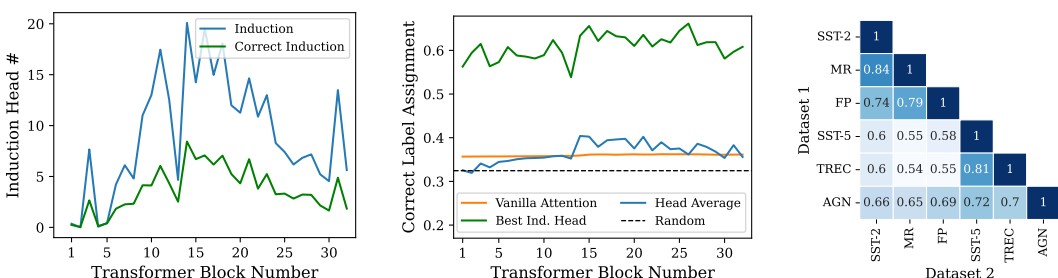

Figure 39: Augmentated results towards Fig. 6 on Llama 3 8B.

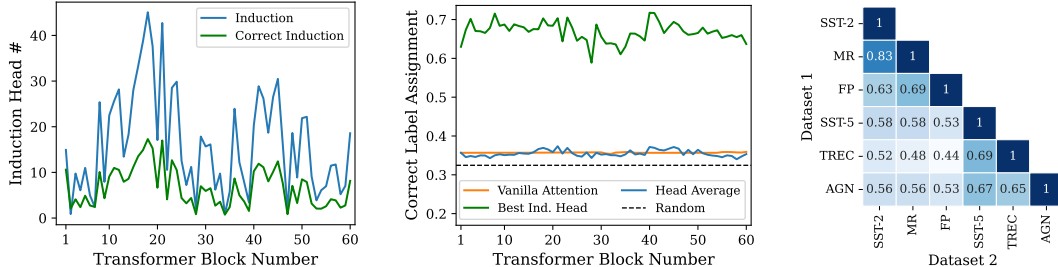

Figure 40: Augmentated results towards Fig. 6 on Falcon 40B.

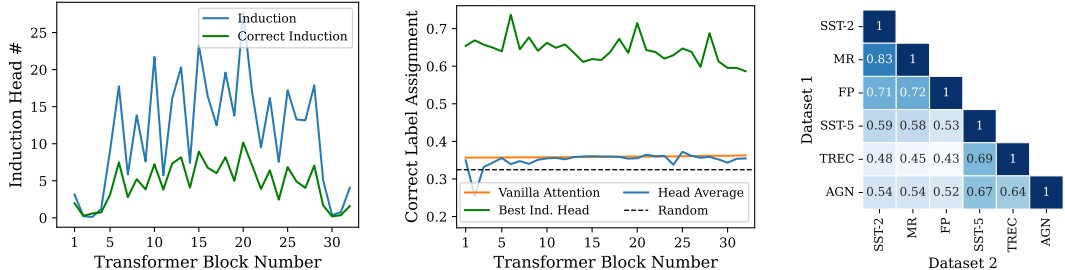

Figure 41: Augmentated results towards Fig. 6 on Falcon 7B.

