# OpenReview forum: "Revisiting In-context Learning Inference Circuit in Large Language Models"
_ICLR.cc/2025/Conference — ICLR 2025 Poster_

### Official Review · Reviewer_yFjP · 2024-10-28

**Soundness:** 3
**Presentation:** 4
**Contribution:** 3
**Rating:** 8
**Confidence:** 3

**Summary:**

This paper proposes a three-stage ICL circuit hypothesis and provides thorough empirical examinations of the existence and significance of these stages. Within this circuit framework, many phenomena are explained, such as how Forerunner Tokens encode input text representations and the bias in input text encoding towards position. These findings present intriguing insights.

**Strengths:**

- **Originality:** To my knowledge, the three-stage circuit proposed by the authors is a novel contribution.
- **Quality:** The hypothesis put forward is reasonable, and the experiments are thorough with a well-crafted methodology.
- **Clarity:** The arguments and evidence presented in the paper are clear, and the experimental descriptions are appropriately detailed.
- **Significance:** Currently, ICL is one of the most important applications in the LLM field, and understanding the mechanisms behind ICL will greatly aid in enhancing its performance.

**Weaknesses:**

The three-stage ICL framework appears to have implicit applicability conditions, which I believe should be clarified.

For example, in Fig. 1 on page 2, a few-shot scenario with $ k=2 $ is presented, which indeed fits the three-stage ICL circuit framework. However, in a zero-shot scenario ($ k=0 $), step 1 may still exist, but steps 2 and 3 would not be applicable. In a few-shot scenario with $ k=1 $, steps 1 and 2 might still apply, but step 3 cannot exist.

Therefore, the framework proposed in this paper should be limited to discussions of scenarios where $ k \geq 2 $. A related question arises: if the focus is restricted to this scenario, what potential issues might emerge?

Furthermore, if we condition on $ k \geq C $ (where $ C $ is a fixed value), could this value vary depending on the problem type? For instance, in tasks like SST-2 and SST-5, which have different label set sizes, might the value of $ C $ differ across these scenarios?

**Questions:**

The three-stage framework proposed in the paper is quite interesting. The questions I have raised are mentioned in the weaknesses section. Here, I would like to know what inspired you to propose this framework. Each part of the framework consists of very specific ideas—were they derived from repeated trial and error, or were they inspired by something else? Alternatively, are they improvements on a significant prior work?

---

> ### Author Response · Authors · 2024-11-14
>
> Dear Reviewer yFjP,
>
> We sincerely appreciate your creative and professional review comments, as well as your positive feedback on our work. Your insights have been invaluable to us, and we are grateful for the opportunity to address your questions and refine our paper based on your suggestions:
>
> ----
>
> **Reply to weakness:**
>
> **Q: Inference dynamics when the label corresponding to the query is not present in the context.**
>
> *Clarified: Revision 5*
>
> A: Thank you for your thought-provoking question in the weakness! Please allow us to summarize your concern as “Whether the circuit fails when the ground-truth label of the query does not appear in the context”?
>
> As background, we would like to clarify one point: in some studies, this situation is typically considered an exception, referred to as In-weight Learning [1,2,3]. However, we believe that our framework remains expressive in such cases:
>
> In this situation, the induction head in step 3 struggles to retrieve similar information from the context, meaning it cannot add label-related information to the query. However, it is still operational, albeit without making a positive contribution. Therefore, although all inference steps are functioning at this point, only step 1 actively contributes. Furthermore, when step 3 is unable to function, we do not actually need to be concerned about the step 2. Therefore, this type of in-weight learning scenario (even when some demonstrations are provided) aligns with the zero-shot case, where only the step 1 is active, and the model relies solely on the information within its internal parameters to solve the problem.
>
> However, please kindly note that our inference circuit still retains on such scenarios, so we have not imposed specific conditions on the demonstrations given.
>
> Thank you for your insightful question. We will include a brief section in our upcoming revision to clarify this point.
>
> ----
>
> **Reply to questions:**
>
> **Q: The source of inspiration for our work.**
>
> A: Regarding your inquiry about the inspiration behind our work, in brief, some previous studies ([1, 2], et al.) have utilized highly embedded inputs (e.g., image features processed through ResNet) as the inputs (namely, $x$ in our paper) to study the ICL process and have obtained intriguing results. However, these results cannot be directly applied to large language models, as the inputs of LLMs are typically not highly embedded. This motivated us to think: if we consider the first few layers of a language model as analogous to these embedding processors (such as the ResNet), and the hidden state corresponding to the last token of the input text (the forerunner token in the paper) as analogous to these embedded features, we could bridge the gap between prior research and LLMs. This paper is the outcome of such bridging, and corresponding to the previous observation about some inference phenomenon (mentioned in 5.3 of our paper), we also conducted detailed measurements to uncover more characteristics of the inference process, which we found to be both novel and intriguing.
>
> ----
>
> **Coming Revisions (during the author-response period):**
>
> As you kindly mentioned in the weakness, we are going to clarify and show how the circuit acts when the ground-truth label is not in the context. We will add a brief explanation to clarify this point.
>
> ----
>
> We are grateful for your detailed comments and the time you have taken to help us refine our study. Thank you again!
>
> **Reference**
>
> [1]. Chan, et al., Data Distributional Properties Drive Emergent In-Context Learning in Transformers. NeurIPS 2022.
>
> [2]. Gautam Reddy. The Mechanistic Basis of Data Dependence and Abrupt Learning in an In-context Classification Task. ICLR 2024.
>
> [3]. Chan, Bryan, et al. "Toward Understanding In-context vs. In-weight Learning." arXiv:2410.23042. 2024.

---

> > ### Comment · Reviewer_yFjP · 2024-11-15
> > **Reply**
> >
> > Thank the authors.
> >
> > So, what you're saying is that Situation 1 fully aligns with your framework, while Situation 2 also aligns, but may introduce in-weight learning, leading to a slightly lower degree of alignment, right?
> >
> > **Situation 1**
> >
> > ```text
> > Taylor Swift: Singer
> > Donald Trump: Politician
> > Ilya Sutskever: Researcher
> > Geoffrey Hinton:
> > ```
> >
> > **Situation 2**
> >
> > ```text
> > Taylor Swift: Singer
> > Donald Trump: Politician
> > Ilya Sutskever: Researcher
> > Michael Jordan:
> > ```
> >
> > I find your work very interesting, and the experiments and reasoning are solid. However, I think you should ensure that your framework has strong generalizability. If there are any ambiguities (such as the "albeit without making a positive contribution" point you mentioned), it may reduce the credibility of the framework and its generalizability. I would like to see stronger clarifications on these aspects.

---

> ### Author Response · Authors · 2024-11-19
>
> Thank you for your prompt response and the thorough explanation of the question. We fully agree on the importance of ensuring the generalizability of our work.
>
> Regarding the scenario you raised in Situation 1, which represents a standard in-context learning (ICL) setting where the ground-truth label is provided within the context, we are confident that our framework generalizes effectively since our extensive experiments, including detailed ablation studies, support this assertion and underscore the robustness of our approach in such scenarios.
>
> For Situation 2, we conducted additional experimental investigations and noted that certain inconsistencies could arise with our proposed framework. Specifically, while Steps 1 and 2 maintain consistency due to their independence from the query, Step 3—involving the induction head—behaves differently because similar features are not accessible. This difference highlights a limitation in our framework under these conditions.
>
> We would like to clarify this in the limitation section and show our experiment results in the Appendix.
>
> Nevertheless, we wish to stress that the main contributions and significance of our paper remain robust. Situation 2 is often associated with in-weight learning, which prior research has identified as fundamentally distinct from, and sometimes even antagonistic to, in-context learning scenarios (as detailed in the referenced literature of the last reply), and some small models are even difficult to answer to both inputs simultaneously with a same set of parameters. Therefore, although Situations 1 and 2 may appear similar in terms of input format, discussing them separately is reasonable and necessary.
>
> We greatly appreciate your careful review and constructive feedback, which have undoubtedly strengthened our work. We welcome any further insights you may have.

---

> ### Author Response · Authors · 2024-11-21
>
> Dear Reviewer yFjP,
>
> Thank you once again for your valuable feedback on our paper. We have tried to address the comment and submitted a revised version for your review, please refer to the global comment "Revision Notes".
>
> - **Q: The condition of this paper.** Clarified in the revision 5.
>
> Please kindly let us know if any further clarification is needed.

---

> ### Author Response · Authors · 2024-12-03
> **More Ablation Experiment in IWL Scenarios (1/2)**
>
> Dear Reviewer yFjP,
>
> Thank you once again for your valuable feedback on our paper.
>
> Regarding your concern, we conduct more experiments to test the generalizability of our conclusion towards scenarios when the ground-truth label does not appear in the context (please allow me to use the acronym IWL to refer to this situation). We find these results to be very interesting, addressing some of your concerns and enhancing the existing conclusions of our paper: generally speaking, these new experiments confirm that: **our theoretical framework is still able to explain inference phenomena in IWL scenarios, thus strengthening our framework.**
>
> In detail, we repeat the ablation study shown in Table 1, but with filtered inputs with no ground-truth query label presented in the context. The results are shown in the table in the following reply, with the same format as Table 1, but the baseline value of random disconnection is omitted for conciseness.
>
> Compared to the results of unfiltered inputs (Table 1), we highlight 4 major phenomena, and explain them within our framework:
> 1. The baseline accuracies (line 1 in the new tables) in IWL settings are lower than in normal settings, even worse than random prediction. This is easy to understand given the frequency bias introduced in Sec. 5.3.
> 2. **As we gradually suppress the induction heads (line 5), the accuracies increase, which is consistent with the findings of induction heads in this paper: in IWL setting, induction heads can not find and copy any correct label-related information in the context, and only copy the label information presented in the context noisily.** Therefore, after removing it from the inference process, the accuracies increase and approach a 0-shot level.
> 3. Interestingly, removing the copying processing (line 4) unexpectedly enhances the accuracy. We infer that the input texts of different labels under the same task still have a certain background-value similarity of the encoding from Step 1 at the forerunner token. Such similarities are propagated into the label tokens by Step 2, enhancing the attention flow from labels to the query's forerunner token in Step 3, causing a decrease in accuracy.
> 4. Removing the demonstration text encoding (line 2) decreases the accuracy to around $0$. According to our discussion about Step 1, in this case, forerunner tokens cannot receive information from the demonstration text, leading an equivalent sequence for the later layers without demonstration text like $[s_1][y_1][s_2][y_2]\dots[s_k][y_k][x_q][s_q][y_q]$. Given that the forerunner token $[s_\cdot]$ is consistent throughout the input, later layers are likely to conduct induction only on the forerunner token, whose subsequent tokens do not contain the correct label, leading to a $0$ accuracy. This is also a critical rebuttal to previous work [1], which believes the label tokens directly collect the information of input text: if that is the case, then disconnecting the links from demonstration texts to the forerunner tokens will not have such a significant disturbance on the accuracy.
>
>
> In summary, these results indicate: **even in the IWL setting, this paper's conclusions can explain the inference phenomena, and the findings in our paper are also significantly enhanced**. This extends the applicability of our conclusions, while, given the existence of some counterexamples shown in the current Appendix G, we still hope that future work can provide a more detailed discussion of the inference dynamics under the IWL scenario.
>
> **Reference**
>
> [1]. Wang, Lean, et al. "Label words are anchors: An information flow perspective for understanding in-context learning." arXiv preprint arXiv:2305.14160 (2023).

---

> ### Author Response · Authors · 2024-12-03
> **More Ablation Experiment in IWL Scenarios (2/2)**
>
> Table. Accuracy variation (\%) with each inference step ablated on **Upper**: Llama 3 8B, **Lower**: Falcon 7B in **IWL scenarios**. Notations are the same with Table 1, significant comparisons are highlighted with arrows.
>
> Llama 3 8B:
> | # |              Attention Disconnected              |       25%       |       50%       |       75%       |       100%       |
> |:-:|:------------------------------------------------:|:---------------:|:---------------:|:---------------:|:----------------:|
> | 1 |            None (4-shot IWL baseline)            |                 |     $\pm 0$     |   (Acc. $33.43\downarrow$)  |                  |
> |   |        **_- Step 1: Input Text Encode -_**       |                 |                 |                 |                  |
> | 2 | Demo. Texts $x_i$ $\rightarrow$ Forerunner $s_i$ |      $-2.31$      |      $-17.90$     |      $-28.58$     |      $-31.22$      |
> | 3 | Query Texts $x_q$ $\rightarrow$ Forerunner $s_q$ |      $-2.67$      |      $-12.96$     |      $-17.87$    |      $-13.41$      |
> |   |         **_- Step2: Semantics Merge -_**         |                 |                 |                 |                  |
> | 4 | Demo. Forerunner $s_i$ $\rightarrow$ Label $y_i$ |      $-1.20$      | $+0.13\uparrow$ | $+0.52\uparrow$ |  $+0.88\uparrow$ |
> |   |     **_- Step3: Feature Retrieval & Copy -_**    |                 |                 |                 |                  |
> | 5 | Label $y_i$ $\rightarrow$ Query Forerunner $s_q$ | $+2.08\uparrow$ | $+2.05\uparrow$ | $+5.40\uparrow$ | $+22.95\uparrow$ |
> |   |               **_Reference Value_**              |                 |                 |                 |                  |
> | 6 |                     Zero-shot                    |                 |      $+17.22$     |   (Acc. $50.65$)  |                  |
> | 7 |                 Random Prediction                |                 |      $-0.93$      |   (Acc. $32.50$)  |                  |
>
> Falcon 7B:
> | # | Attention Disconnected | 25% | 50% | 75% | 100% |
> |:---:|:---:|:---:|:---:|:---:|:---:|
> | 1 | None (4-shot IWL baseline) |  | $\pm 0$ | (Acc. $27.41\downarrow$) |  |
> |  | **_- Step 1: Input Text Encode -_** |  |  |  |  |
> | 2 | Demo. Texts $x_i$ $\rightarrow$ Forerunner $s_i$ | $-17.64$ | $-24.32$ | $-26.17$ | $-26.43$ |
> | 3 | Query Texts $x_q$ $\rightarrow$ Forerunner $s_q$ | $-3.19$ | $-15.33$ | $-21.48$ | $-22.53$ |
> |  | **_- Step2: Semantics Merge -_** |  |  |  |  |
> | 4 | Demo. Forerunner $s_i$ $\rightarrow$ Label $y_i$ | $+2.93\uparrow$ | $+1.10\uparrow$ | $+6.38\uparrow$ | $+5.86\uparrow$ |
> |  | **_- Step3: Feature Retrieval & Copy -_** |  |  |  |  |
> | 5 | Label $y_i$ $\rightarrow$ Query Forerunner $s_q$ | $+8.20\uparrow$ | $+5.63\uparrow$ | $+9.28\uparrow$ | $+29.26\uparrow$ |
> |  | **_Reference Value_** |  |  |  |  |
> | 6 | Zero-shot |  | $+33.58$ | (Acc. $60.99$) |  |
> | 7 | Random Prediction |  | $+5.09$ | (Acc. $32.50$) |  |

---

### Official Review · Reviewer_gt4r · 2024-10-30

**Soundness:** 3
**Presentation:** 4
**Contribution:** 3
**Rating:** 6
**Confidence:** 3

**Summary:**

This paper investigates the mechanisms within large language models (LLMs) that enable in-context learning (ICL) tasks, breaking down the process into three distinct stages: summarization, semantic merging, and feature retrieval/copying. The study employs a variety of experiments across multiple LLMs to validate its findings. Overall, this paper presents valuable insights that can contribute to the field of LLM research, particularly within the ICL community.

**Strengths:**

1. The findings in this paper are clearly explained. Experimental results and visualizations enhance readability and help the audience follow the study's progression easily.

2. This work is well-connected to existing ICL research, with discussions that compare its findings to prior studies on ICL explainability and demonstration selection.

3. The study uses multiple LLMs, strengthening the generalizability of the findings across different model architectures.

4. The insights provided are thought-provoking and have potential practical implications for ICL applications.

**Weaknesses:**

1. [Section 3.1] The authors used mutual nearest-neighbor kernel alignment to evaluate LLMs' summarization abilities. However, the term “summarize” lacks clarity. Does it refer to encoding capabilities similar to those in BGE?

2. [Section 3.1] Additionally, the kernel alignment metric may not be sufficiently robust, as alignment scores in Figure 2 range only from 0.25 to 0.35, which is not significant enough. Consequently, the finding on “summarization” may hold only to a limited extent.

3. [Section 4.1 – Copying from Text Feature to Label Token] It is unclear whether the copying mechanism is applied solely to label tokens or if it extends to other tokens within the input. Using results from other tokens as a baseline could provide a more nuanced understanding of the copying process.

4. [Figure 5, Right] After layer 40, the classification accuracy drops significantly. The authors did not investigate potential reasons for this decline. Could it be due to the gradual degradation of copied information?

5. The experimental setup in Section 5.1 is insufficiently detailed. For instance, how many attention heads are disconnected at each layer? Additionally, the experiments lack certain baselines, such as randomly disconnecting some attention heads to observe the impact on model performance.

Despite these questions and weaknesses, I believe this paper still offers meaningful insights.

**Questions:**

Please see weaknesses.

**Details Of Ethics Concerns:**

N.A.

---

> ### Author Response · Authors · 2024-11-14
>
> Dear Reviewer gt4r,
>
> We truly appreciate the time and attention you devoted to reviewing our paper. Your thoughtful and thorough feedback has provided us with important guidance for revising our paper, and also inspiring new research possibilities. Below, we provide our responses to the points you have kindly raised.
>
> ----
>
> **Reply to weakness:**
>
> **Q1: The term “summarize” lacks clarity.**
>
> *Revised: Revision 8*
>
> A1: Yes, the term “summarize” refers to encoding capabilities, where LMs encode the information of input text $x$ into its forerunner token $s$. Could you please specify how this term lacks clarity (which concepts are not conveyed / with which concepts might be confused?) to help us choose a better term in our revision?
>
> **Q2: The significance of “summarization”.**
>
> A2: Our random baseline is 0.125, and the results we have obtained are twice that value, which we believe suggests the presence of this encoding process. Furthermore, in 3.2, we have demonstrated that the inner encoding can be successfully leveraged by a linear classifier, yielding high accuracy on downstream tasks. This further supports the reliability of the evidence.
>
> **Q3: The scope of the copy processing.**
>
> *Revised: Revision 3*
>
> A3: Thank you for raising such an interesting question. Our current intuition is that the copying mechanism is widespread among tokens and does not show specific positional selectivity. However, we acknowledge that this intuition requires experimental validation, and we plan to include both this conclusion and the corresponding experiments in our next revision during the author-response period.
>
> **Q4: The reason for accuracy dropping in late layers in Fig. 5 Right.**
>
> *Revised: Revision 7*
>
> A4. Thank you for your thought-provoking question again. Based on our experimental results, we believe we can support the point you kindly raised, “it is due to gradual degradation of copied information”. This is evident from the following observations: (1). the accuracy drops from layer 37 as shown in Fig. 5 Right; (2). the copying process in the second step does not persist into the deeper layers, as shown in Fig. 5 Left. We will include a brief description of this process in our revision.
>
> Additionally, we believe there is a profound principle underlying this observation: why are these copied features erased, even with the presence of residual connections? This leads to a new research question: how do language models forget or update information within hidden states to maintain their hidden states capacity? While these questions extend beyond the scope of this paper, we believe they can inspire a more in-depth series of studies. We sincerely appreciate your insightful comments.
>
> **Q5: The setting of ablation experiments and the reference value.**
>
> *Emphasized: Revision 1, 2*
>
> A5: First, we would like to clarify the setup of the ablation experiment: we attribute each step of the inference process to specific attention connections, which are shown in Table 1. For instance, for step 1, we attribute it to the attention connections from the input text tokens to the forerunner tokens. When we aim to remove this step from the inference, we eliminate all corresponding attention connections from layer 0 to layer $\\{25\\%, 50\\%, 75\\%, 100\\%\\}\times\mathrm{Total Layers}$.
>
> For the baseline problems, please kindly refer to the small numbers presented beneath the results in Table 1—these are not standard deviations, but the control values when the same amount of randomly selected attention connections are removed. The accuracy drops in the control values are clearly weaker than the experimental values where the important attention connections are removed, leading us to believe our results are reliable.
>
> ----
>
> **Coming Revisions (during the author-response period):**
>
> [To Q3] We are going to measure the scope of the copy processing.
>
> [To Q4] We are going to describe the erasing processing briefly in 4.2.
>
> [To Q5] We are emphasizing the setting of the ablation experiment in maybe Appendix.
>
> Moreover, we acknowledge that many experienced researchers might interpret the small numbers in Table 1 as standard deviations like the usual practice of academic papers, so we are revising the presentation to emphasize this, which will be included in the upcoming revision.
>
> ----
>
> Once again, thank you for your time and valuable feedback. We look forward to submitting the revised manuscript and hope it meets your expectations.

---

> ### Author Response · Authors · 2024-11-21
>
> Dear Reviewer gt4r,
>
> Thank you once again for your valuable feedback on our paper. We have tried to address the comments and submitted a revised version for your review, please refer to the global comment "Revision Notes".
>
> - **Q3: The scope of the copy processing.** Addressed in the revision 3.
> - **Q4: The reason for accuracy dropping in late layers in Fig. 5 Right.** Discussed in the revision 7.
> - **Q5: The setting of ablation experiments and the reference value.** Addressed in the revisions 1 and 2.
>
> Please kindly let us know if any further clarification is needed.

---

> ### Comment · Reviewer_gt4r · 2024-11-22
>
> Thank the authors for their detailed and thoughtful reply. I appreciate the effort in addressing my concerns, and most of them have been adequately resolved.
>
> Regarding Q1, I believe the term "encoding capabilities" effectively conveys the meaning of "summarize". My suggestion is for the authors to **explicitly clarify that "summarize" refers to "encoding capabilities" in the paper**. Because while reading the paper, I noticed both terms—"summarize" and "encoding capabilities"—are used, but their relationship is not fully clear. I am not sure whether they are equivalent or if one concept encompasses the other.
>
> Overall, I think it is a good paper, and I am inclined to recommend acceptance.

---

> > ### Author Response · Authors · 2024-11-22
> >
> > Thank you very much for your thoughtful feedback and for appreciating our efforts in addressing your concerns. We are grateful for your constructive comments and your positive view of our work.
> >
> > Regarding your suggestion, we understand your concern about the clarity between the terms "summarize" and "encoding capabilities." To address this, **we have revised the terminology in the manuscript to rename "summarize" with "Input Text Encode" to enhance clarity and ensure consistency.**
> >
> > We hope this revision resolves your concern and improves the overall readability of the paper. Thank you again for your valuable suggestions and for recommending our work for acceptance.

---

### Official Review · Reviewer_Z26B · 2024-11-04

**Soundness:** 3
**Presentation:** 2
**Contribution:** 2
**Rating:** 6
**Confidence:** 3

**Summary:**

The authors propose a three-step inference circuit to capture the in-context learning (ICL) process in large language models (LLMs):

1. Summarize: Each input (both demonstration and query) is encoded into linear representations in the model's hidden states.

2. Semantics Merge: The encoded representation of each demonstration is merged with its label, creating a joint representation for the label and demonstration.

3. Feature Retrieval and Copy: The model retrieves and copies the label representation most similar to the query's representation, using this merged representation to predict the query's label.

This circuit explains various ICL phenomena, such as position bias, robustness to noisy labels, and demonstration saturation. Ablation studies show that removing steps in this process significantly reduces performance, supporting the dominance of this circuit in ICL. The paper also identifies some bypass mechanisms that operate in parallel, assisting the inference circuit through residual connections.

**Strengths:**

1. The authors proposed to use the mutual nearest-neighbor kernel alignment of the intermediate representations of LLMs and sentence embeddings produced by another pre-training model to assess the quality of these representations. This method is novel.

2. Extensive analysis has been performed on all three steps of the proposed framework. Possible explanations have also been provided for many phenomena.

3. The experiments are performed with real-world LLMs and datasets, which makes the insights more likely to be useful in practice.

4. The paper is well-written and easy to follow.

**Weaknesses:**

1. The majority of the analysis is based on associations without verifying their strength and whether those effects are causal. For example, Figure 2 right does not look significant enough for me. The peaks highlighted in Figure 5 also look pretty noisy to me.

2. The causal evidence that the authors provided in the ablation study only shows the effect of deleting the hypothesized important components in ICL. What if unimportant components are deleted? Would they have a similar effect? Only if the unimportant components have a significantly weaker effect on ICL performance, can we draw a causal conclusion that the proposed three-step process dominates ICL.

**Questions:**

See weaknesses.

---

> ### Author Response · Authors · 2024-11-14
>
> Dear Reviewer Z26B,
>
> We sincerely appreciate your thorough review and valuable feedback on our manuscript. Your insightful comments not only enhance our research but also prompt further research ideas. Below, we provide our responses to the points you have kindly raised.
>
> ----
>
> **Reply to questions:**
>
> We sincerely appreciate the valuable questions you raised, and believe that the two questions you kindly pointed out pertain to the causality of our proposed inference framework.
>
> **Q1: Lack of baseline in ablation studies.**
>
> *Emphasized: Revision 1*
>
> A1: First, we would like to address your **second** question regarding the importance of referencing the effect of removing unimportant components in the ablation study. Please kindly refer to the small numbers presented beneath the results in Table 1—these are not standard deviations, but the control values when the same amount of randomly selected attention connections are removed. The accuracy drops in the control values are clearly weaker than the experimental values where the important attention connections are removed, leading us to believe our results are reliable.
>
> **Q2: How significant / associated is this inference circuit?**
>
> *Clarified: Revision 4*
>
> A2: As for your **first** question, to our knowledge, measuring how various modules in a Transformer are *actually* connected remains an open question. Although some idealized concepts are provided in the literature (please refer to the Virtual Weights section in [1]) to calculate the similarity of the reading space (defined by attention head’s $W_Q$, $W_K$ and $W_V$) and the writing space ($W_O$), they are difficult to apply to large language models as they overlook numerous modules (e.g., LayerNorm), which can deform the hidden space. Therefore, our paper uses the ablation study in Table 1 to confirm that our inference framework is effective and dominant.
>
> > *When we say “actually”, we want to evaluate whether two modules are sharing a common subspace of hidden states to communicate, as opposed to simply being sequentially “connected” by the layer order.
>
> ----
>
> **Coming Revisions (during the author-response period):**
>
> [To Q1] We acknowledge that many experienced researchers might interpret the small numbers in Table 1 as standard deviations like the usual practice of academic papers, so we are revising the presentation to emphasize this, which will be included in the upcoming revision.
>
> ----
>
> Once again, we are deeply grateful for your professional review and constructive suggestions.
>
> **Reference**
>
> [1]. Elhage, Nelson, et al. "A mathematical framework for transformer circuits." Transformer Circuits Thread 1.1 (2021): 12.

---

> ### Author Response · Authors · 2024-11-21
>
> Dear Reviewer Z26B,
>
> Thank you once again for your valuable feedback on our paper. We have tried to address the comments and submitted a revised version for your review, please refer to the global comment "Revision Notes".
>
> - **Q1: Lack of baseline in ablation studies.** Addressed in the revision 1.
> - **Q2: How significant / associated is this inference circuit?** Discussed in the revision 4.
>
> Please kindly let us know if any further clarification is needed.

---

### Official Review · Reviewer_keS9 · 2024-11-10

**Soundness:** 3
**Presentation:** 2
**Contribution:** 2
**Rating:** 6
**Confidence:** 3

**Summary:**

This paper aims to explain the mechanisms behind in-context learning (ICL) using the inference circuit framework.

According to the authors, the ICL process consists of three internal steps:
1. Summarize: Large language models (LLMs) encode each demonstration within its corresponding forerunner token,  $s_i$ .
2. Semantics Merge: The semantics of each demonstration and its label are combined into the representation of the label  $y_i$ .
3. Feature Retrieval and Copy: LLMs rely on the accumulated labels  $y_{1:k}$  to respond accurately to the query  $s_q$ , yielding the most appropriate answer.

Each step is empirically validated using methods such as kernel alignment and embedding comparisons. The authors also seek to align their findings with those of prior research, reinforcing the credibility of the arguments presented in this work.

**Strengths:**

- Attempts to explain the inner workings of ICL, based on reasonable assumptions and investigative tools.
- The findings align with previous work, encouraging readers to accept the claims presented in the paper.
- Visualized results help readers quickly grasp the core concepts and findings of the paper.

**Weaknesses:**

- While the proposed framework is logical and reasonable, it remains challenging to argue definitively that the core mechanism of ICL follows the assumptions presented in the paper. As noted in Section 5.2, there are exceptions that do not align well with the proposed framework, raising concerns that the explanations may be superficial and fail to capture the essence of ICL. This is understandable, as fully explaining the inner workings of neural networks is inherently difficult, if not nearly impossible.
- I am somewhat unclear about the core novelty of this paper. As I understand it, the primary contribution seems to be the attempt to apply the existing inference circuit framework to the ICL of specific LLMs, including LLaMA 3. In Section 2.1, I did not find explanations that clarify why the procedure conducted in this paper is particularly innovative, compelling, or novel. More comprehensive comparisons with prior work employing induction or inference circuits to illustrate the inner workings of ICL would be helpful to underscore the merits and uniqueness of this study.
- In Section 3.1, sentence embeddings generated by an external encoder (BGE M3) are compared to hidden representations computed by an LLM. Since these two representations come from different models, without any modification or fine-tuning, there is a risk that their vector spaces are not aligned or compatible. This raises concerns about whether this experiment is sufficiently reasonable.

**Questions:**

Please refer to the Weaknesses section.

---

> ### Author Response · Authors · 2024-11-14
>
> Dear Reviewer keS9,
>
> Thank you for your insightful review of our manuscript. We greatly appreciate the thoroughness of your comments and the helpful suggestions, which will certainly strengthen our paper. Below, we provide our responses to the points you have kindly raised.
>
> ----
>
> **Reply to questions:**
>
> **Q1: Exceptions of inference behavior.**
>
> A1: Firstly, in the ablation experiment described in 5.1, when we completely suppressed the proposed inference framework, the accuracy dropped to a random level, suggesting that the proposed inference framework might be the dominant mechanism.
>
> Even with exceptions (which we refer to as bypasses), we believe that our paper's contribution remains: LLMs are trained on natural datasets by gradient descent, during which numerous inference behaviors may emerge. The circuits we have identified highlight a significant pattern among these noisy behaviors, which we believe is sufficiently impactful. We fully agree with your statement in the latter part: it is nearly impossible to enumerate all inference pipelines and their interactions, but our paper at least uncovers one of the most significant patterns.
>
> **Q2: Novelty.**
>
> *Clarified: Revision 6*
>
> A2: The main novelty of our paper lies in adapting and conducting detailed measurements of inference circuits, originally discovered in toy models, within real, large-scale language models. This has provided numerous insights and successfully explained a range of observed inference phenomena with a single circuit. In our view, please allow us to state that the unification of these intricate inference phenomena (as summarized in Section 5.3 of the paper) into a single inference circuit is both novel and exciting.
>
> **Q3: Robustness of Kernel Alignment.**
>
> A3: Your insight is absolutely correct: in fact, the feature vectors generated by BGE and those produced by LLM are highly likely to exist in different feature spaces, which is why we use Kernel Alignment to test the similarity between these feature vectors. The reason is that Kernel Alignment measures the similarity of the neighborhood relationships within the feature sets derived from the same set of objects by different models, rather than directly computing the similarity of feature vectors from different spaces. This makes sense: our long-standing practice starting from word embeddings tells us that adjacency relationships determine semantics. Therefore, even if the feature vectors come from different models, Kernel Alignment can still stably compute their semantic similarity, as also shown in previous work [1].
>
> Additionally, in 3.2, we have strengthened the results of Kernel Alignment: we have confirmed that the encodings on the forerunner tokens can be used for linear classifiers and achieve high classification accuracy, which means the features collected enough information from the input text.
>
> ----
>
> **Coming Revisions (during the author-response period):**
>
> [To Q2] We will emphasize the novelty of our work, perhaps in the discussion section.
>
> ----
>
> We truly appreciate your thoughtful review and will incorporate your suggestions to improve the paper. Thank you again for your support.
>
> **Reference**
>
> [1] Huh, Minyoung, et al. "Position: The Platonic Representation Hypothesis." ICML 2024.

---

> ### Author Response · Authors · 2024-11-21
>
> Dear Reviewer keS9,
>
> Thank you once again for your valuable feedback on our paper. We have tried to address the comments and submitted a revised version for your review, please refer to the global comment "Revision Notes".
>
> - **Q2: Novelty.** Addressed in the revision 6.
>
> Please kindly let us know if any further clarification is needed.

---

### Author Response · Authors · 2024-11-21
**Revision Notes**

Dear AC and Reviewers,

Thanks for your valuable time in improving our paper. We have tried to address the review comments and submitted a revised version.

Note: Due to the inclusion of new content, we have extended the main body to a 10th page and reformatted the manuscript for better organization. **While this may appear as an extensive change in some PDF comparison tools, we assure you that no content beyond what is explicitly mentioned in these notes has been altered.**


**Revisions According to Review**

(Revised on November 21, 2024)

1. **[Reviewer Z26B Q2, Reviewer gt4r Q5] The baseline value of ablation experiment.**


   *Revision: Reformatting Table 1, 2, 6*


   The captions and small numbers in Tables 1, 2, and 6 have been highlighted with color to clarify that they represent baseline results. Additionally, we have included the standard deviation data for these baseline results, ensuring they no longer appear as the standard deviation of a main result.

2. **[Reviewer gt4r Q5] The detailed settings of ablation experiments.**


   *Revision: Experiment setting added in Appendix A.7*

   Appendix A.7 has been added to provide detailed information on the ablation experiment settings.


3. **[Reviewer gt4r Q3] The token selectivity of the copy processing.**


   *Revision: Additional context in Sec 4.1 and Appendix D*


   Our experiments demonstrate that the copying mechanism is inherent to the model and extends to other tokens than the label tokens. These findings are now referenced in Section 4.1 (“Text Representations are Copied without Selectivity”) and detailed in Appendix D.


4. **[Reviewer Z26B Q1] Limitation about the connectivity of attention head.**


   *Revision: Limitation Section*


   Limitation 1 has been expanded to discuss future work focusing on measuring the connection magnitude of model components.


5. **[Reviewer yFjP] Limitation about the application condition.**


   *Revision: Limitation Section, Appendix G*

   The original Limitation 2 has been merged into Appendix B.1, and a new Limitation 2 has been added to address Reviewer yFjP's concerns about the application conditions of this work. Further details are provided in the newly added Appendix G.


6. **[Reviewer keS9 Q2] Relationship with Previous Works.**


   *Revision: Additional context “Comparison with Previous Works” in Discussion*

   We have added comparisons with previous interpretability research to highlight the novelty and contributions of our work.


7. **[Reviewer gt4r Q4] The reason for accuracy dropping in late layers in Fig. 5 Right.**

   *Revision: Additional context at the end of Sec. 4.1*

   *(November 27) Replaced at the end of the paragraph: "Hidden States of Label Tokens are Joint Representations of Text Encodings and Label Semantics."*

   We have added a brief description of the accuracy decline, which is aligned with our findings of Fig. 5 Middle.

----

(Revised on November 22, 2024)

8. **[Reviewer gt4r Q1] Term "Summarize" lacks clarity.**

   *Revision: Rename the "Summarize" to "Input Text Encode"*

   We have renamed the "Summarize" to "Input Text Encode" throughout the whole paper, and some expressions have been revised to fit this.

**Other Minor Revisions**


- The Reproducibility Statement / Author Contributions (reserved) / Acknowledgments (reserved) have been moved to the end of the main body. (Revised on November 21)
- Some figures and tables have been moved / resized to make better use of the available space. (Revised on November 21, 27)
- Minor word-level fixes (e.g., typos and grammatical errors) have been made without affecting the semantical content. (Revised on November 21, 22, 27)

We believe we have carefully revised our paper to address the key concerns raised in the reviews, and issues not reflected in the revisions have been discussed in our response in detail. Should there be any further questions or points requiring clarification, we are more than willing to engage in additional discussions.

If you have any other concerns, please kindly let us know.

---

### Meta-Review · Area_Chair_Aouk · 2024-12-19

**Metareview:**

**Summary:**
The paper investigates the mechanisms behind In-Context Learning in large language models. The authors introduce a three-step inference circuit comprising (1) input text encoding, (2) semantic merge, and (3) feature retrieval and copying. Through empirical analysis, they demonstrate that this circuit effectively captures observed phenomena like positional bias, noise robustness, and demonstration saturation in ICL. Ablation studies reveal the circuit's dominance in driving ICL while identifying auxiliary bypass mechanisms enabled by residual connections.

**Strength:**
- Explains the inner workings of In-Context Learning (ICL) based on reasonable assumptions and robust investigative tools.
- Extensively analyzes all three steps of the proposed three-stage inference circuit.
- Conducts experiments on real-world LLMs and datasets, ensuring the insights are applicable in practice.

**Weakness:**
- While the proposed framework is logical, it is challenging to argue definitively that the core mechanism of ICL aligns with the assumptions presented.
- Limited clarification why the procedure conducted in this paper is particularly innovative, compelling, or novel.

**Additional Comments On Reviewer Discussion:**

The authors have provided thorough responses to the various comments raised by the reviewers, but unfortunately, most reviewers did not engage further. From my assessment, the concerns raised have been adequately addressed, and the revised paper shows significant improvement compared to the original version. Additionally, as the initial scores were already leaning towards acceptance, I also recommend accepting the paper.

---

### Decision · Program_Chairs · 2025-01-22

Accept (Poster)